# Identification of wind turbine main shaft torsional loads from high-frequency SCADA measurements using an inverse problem approach

W. Dheelibun Remigius[1] and Anand Natarajan[1]

[1]Department of Wind Energy, Technical University of Denmark, Frederiksborgvej 399, 4000 Roskilde, Denmark.

**Correspondence:** W. Dheelibun Remigius (drwp@dtu.dk)

**Abstract.** To assess the structural health and remaining useful life of wind turbines within wind farms, the site-specific structural response and modal parameters of the primary structures are required. In this regard, a novel inverse problem-based methodology is proposed here to identify the dynamic quantities of the drivetrain main shaft, *i.e.,* torsional displacement and coupled stiffness. As a model-based approach, an inverse problem of a mathematical model concerning the coupled shaft torsional dynamics with high-frequency SCADA measurements as input is solved. It involves Tikhonov regularisation to minimize the measurement noise and irregularities on the shaft torsional displacement obtained from measured rotor and generator speed. Subsequently, the regularised torsional displacement along with necessary SCADA measurements is used as an input to the mathematical model and a model-based system identification method called the Collage method is employed to estimate the coupled torsional stiffness. It is also demonstrated that the estimated shaft torsional displacement and coupled stiffness can be used to identify the site-specific main shaft torsional loads. It is shown that the torsional loads estimated by the proposed methodology is in good agreement with the aeroelastic simulations of the Vestas V52 wind turbine. Upon successful verification, the proposed methodology is applied to the V52 turbine to identify the site-specific main shaft torsional loads and damage equivalent load. Since the proposed methodology does not require a design basis or additional measurement sensors, it can be directly applied to wind turbines within a wind farm that possess high-frequency SCADA measurements.

## 1 Introduction

Monitoring of wind turbines within wind farms is increasingly becoming very important due to the need to detect anomalous behaviour, plan inspections or preventive maintenance, and compute the remaining useful life of specific structures. The site-specific structural dynamic quantities such as structural response and modal parameters could assist in the condition monitoring of wind turbines.

The structural response of the wind turbines are measured using load instrumentation such as accelerometers (Koukoura et al., 2015; Pahn et al., 2017; Norén-Cosgriff and Kaynia, 2021) and an output-based operational modal analysis (OMA) (Wang et al., 2016) technique is employed for the identification of the modal parameters. OMA methods can be broadly classified into two categories: (i) Time domain-based methods, and (ii) Frequency domain-based methods (Zhang et al., 2010). A recent comprehensive review on various time domain and frequency domain-based OMA methods can be found in Zahid et al.

(2020). Upon identification of the modal parameters together with the measured structural response, an inverse problem-based technique is employed for the estimation of the site-specific loads (Pahn et al., 2017). Alternatively, one can use strain gauge based load sensors to measure the site-specific loads directly (IEC 61400-13).

The addition of new instrumentation to existing turbines, such as the installation of strain gauges and accelerometers can be costly and also require repetitive calibration and synchronization of their measurement signals with the turbine computer. Most

wind farm operators also do not possess the aeroelastic design parameters of their wind turbines and hence cannot simulate the mechanical loads acting on their wind turbines. Monitoring turbine primary structures through existing Supervisory Control and Data Acquisition (SCADA) system-based measurements can allow cost-effectiveness and provide valuable information to the wind farm operator. Usually, such monitoring through SCADA only provides information on power performance and not regarding turbine structural integrity. Since there are two SCADA signals (i.e., rotor speed and generator speed) related to

torsional oscillations of the main shaft of wind turbine drivetrains, the same can be used to quantify the torsional dynamics of the main shaft.

For this purpose, an inverse problem-based approach is developed here to determine the torsional stiffness and response of the main shaft of a wind turbine, using existing high-frequency SCADA measurements such as the rotor speed and generator speed. This is a cost-effective alternative approach that is being proposed for the main shaft without using any additional

measurement sensors or an aeroelastic design basis of the wind turbine. The proposed inverse problem approach is a model-based approach whereby a mathematical model concerning the shaft torsional dynamics will be utilized to obtain both the torsional displacement and the coupled torsional stiffness in a continuous time domain. It involves Tikhonov regularisation (Tikhonov, 1963) for regularising the measurement data and the Collage method (Kunze and Vrscay, 1999) for estimating the torsional stiffness.

The concerned mathematical model comprised of differential equations will be solved for the shaft torsional velocity with high-frequency rotor and generator speed measurements as inputs. Subsequently, the main shaft torsional displacement is obtained by numerically integrating the shaft torsional velocity. However, numerical integration is based on time-marching algorithms, and the lack of initial conditions makes displacement reconstruction an ill-posed problem (Hong et al., 2008). Since it is an ill-posed problem, influence of measurement noise will get amplified during the time marching procedure and

results in an erroneous displacement. This inaccurate displacement also leads to drastic errors in the system modal parameter estimation. Hence, one needs to go for regularisation techniques to smoothen the reconstructed displacement. Hansen (2005) discussed the nature of various ill-posed problems and presented several solution methodologies. Though there are many regularisation techniques available, Tikhonov, truncated singular value decomposition (SVD) and nuclear norm are a few of the popular techniques (Aarden, 2017). Among all these regularisation techniques, Tikhonov regularisation has been widely

used in many engineering applications (Ronasi et al., 2011; Hào and Quyen, 2012; Bangji et al., 2017; Nieminen et al., 2011) and it has been studied extensively in the field of inverse problems (Hansen, 2005) as well. Further, digital filters and frequency domain integration approach (FDIA) are also widely used techniques in the literature to reconstruct displacements from measured accelerations (Hong et al., 2008; Brandt and Brincker, 2014; Qihe, 2019; Lee et al., 2010). However, digital filters such as impulse response filters (IIR) and finite response filters (FIR) have several drawbacks when reconstructing the

low-frequency dominant displacements as is the case here (Lee et al., 2010). On the other hand, the FDIA methods are sensitive to the time interval of the measurements (Lee et al., 2010). It is shown by Lee et al. (2010) that the Tikhonov regularisation is better suited for low-frequency dominant structures. Hence, the same has been employed in the present work. Tikhonov regularization minimizes the error using the least-square criterion and by means of numerical damping, it also minimizes the effect of measurement noise.

Upon obtaining the regularised shaft torsional displacement, the same mathematical model is utilized to obtain the shaft stiffness. For this purpose, the Collage method - a model-based system identification technique - is used (Kunze and Vrscay, 1999; Groetsch, 1993). This method has been successfully applied for the system identification in various differential equations-based problems such as boundary value problems (Kunze et al., 2009), reaction-diffusion problems (Deng et al., 2008) and elliptic problems (Kunze and La Torre, 2016). The Collage method transforms the system identification problem into a minimization problem of a function of several variables (for example unknown system parameters) and then the minimization problem is solved using a minimisation algorithm called the Collage coding (Kunze et al., 2009). Collage coding is a greedy algorithm that attempts to find the approximate solution in a single step without any need for iteration, as is the case for other inverse problem-based methods (Deng and Liao, 2009). Further, the model-based Collage method is simple, easy to implement and computationally inexpensive as compared to the output-based OMA methods.

The estimated shaft torsional stiffness and displacement are further used to identify the site-specific shaft torsional load. This novel methodology can potentially benefit wind farm owners since both the property of the structure in terms of its stiffness and the structural response and the site-specific load can be determined without requiring additional sensors or information from the wind turbine manufacturer. The main shaft torsional load affects the fatigue performance of other drivetrain components such as gearbox and planetary bearings (Dong et al., 2012; Gallego-Calderon and Natarajan, 2015; Ding et al., 2018). Hence, the same site-specific torsional load may be used as an input for quantifying the remaining useful life (RUL) (Ziegler et al., 2018) of the main shaft, gearbox, and other drivetrain components as well. The estimation of RUL/ yearly damage does not require additional historical weather data and condition monitoring data as the wind speed and wind direction measurements are available in the SCADA measurements. However, this is beyond the scope of present work. Further, the proposed approach requires that the sampling frequency of the SCADA measurement be significantly higher than the dominant frequencies of the drivetrain torsional oscillations (i.e., 1p and 3p rotor excitation frequencies and torsional natural frequencies). As a result, the proposed method cannot be used for the turbines that have measurements in terms of 10-min SCADA statistics.

The rest of the paper is organised as follows: the problem formulation consisting of the Tikhonov regularisation and the Collage method is given in section 2; section 3 presents the verification of the proposed formulation; application of the proposed formulation on measurements are presented in section 4.

## 2 Problem formulation

As mentioned in the previous section, the main objective is to identify the shaft torsional displacement and coupled stiffness from SCADA measurements. This is achieved by solving the shaft torsional dynamical equations using a suitable inverse

problem algorithm and the estimated shaft torsional displacement $\boldsymbol{\theta}$ and torsional stiffness $K$ will be utilized for the shaft torsional load estimation. For this purpose, a two-mass model (refer Fig. 1) (Boukhezzar et al., 2007; Girsang et al., 2013;

Berglind et al., 2015) that governs the main shaft torsional dynamics subjected to the rotor and generator torques $\boldsymbol{T}_r$ and $\boldsymbol{T}_g$, respectively, is considered, and the mathematical model is given by Eqs. (1-3). The governing equations are obtained in an inertial frame of reference and converted to the low-speed side of the drivetrain by means of the gear ratio (Boukhezzar et al., 2007). By assuming that the drivetrain components are in a series representation in terms of its modal quantities, effective values for the modal quantities are used in the two-mass model (Girsang et al., 2013).

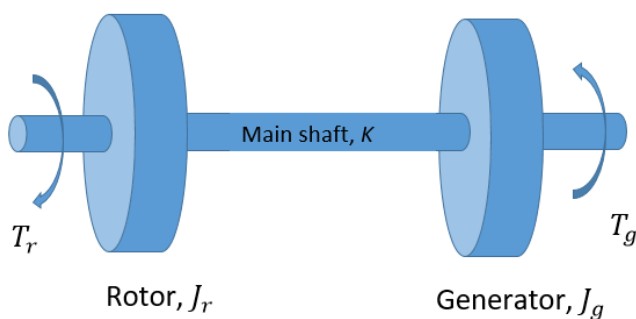

**Figure 1.** A two mass model of wind turbine drivetrain.

$$
\begin{aligned}
\quad J_r \dot{\boldsymbol{\omega}}_r &= \boldsymbol{T}_r - K\boldsymbol{\theta} - C\dot{\boldsymbol{\theta}}, & (1)\\
J_g \dot{\boldsymbol{\omega}}_g &= -\boldsymbol{T}_g + K\boldsymbol{\theta} + C\dot{\boldsymbol{\theta}}, & (2)\\
\dot{\boldsymbol{\theta}} &= \boldsymbol{\omega}_r - \boldsymbol{\omega}_g. & (3)
\end{aligned}
$$

Here, $J_r$ represents the inertia of the rotor, $J_g$ represents the collective inertias of the high-speed shaft (HSS), the gearbox, and the generator, $\boldsymbol{\omega}_r$ and $\boldsymbol{\omega}_g$ are the rotor and generator speeds, respectively, $K$ and $C$ are the effective stiffness and damping of

the drivetrain including the main shaft, HSS and gearbox, and $\boldsymbol{\theta}$ is the torsional displacement of the drivetrain. Throughout the article, all vector quantities have been marked with bold font.

Given the modal parameters $(J_r, J_g, C, K)$ and external torques ($\boldsymbol{T}_r$ and $\boldsymbol{T}_g$), Eqs (1-3) are solved for $\boldsymbol{\omega}_r, \boldsymbol{\omega}_g$ and $\boldsymbol{\theta}$, which is known as a forward problem approach (Pahn, 2013). But given only SCADA measurements, Eqs. (1-3) are solved inversely for $\boldsymbol{\theta}$ and modal parameters. The available SCADA measurements are $\boldsymbol{\omega}_r, \boldsymbol{\omega}_g, P, \beta, \boldsymbol{U}$. Here, $P, \beta, \boldsymbol{U}$ are, respectively, the gen-

erator power, blade pitch angle and wind velocity. The proposed inverse problem approach consists of Tikhonov regularisation for regularising the measurement data and the Collage method for estimating the torsional stiffness and the entire methodology is shown as a flow chart in Fig. 2. In the following, implementation of the Tikhonov regularisation and Collage method on the drivetrain torsional dynamics will be discussed in detail.

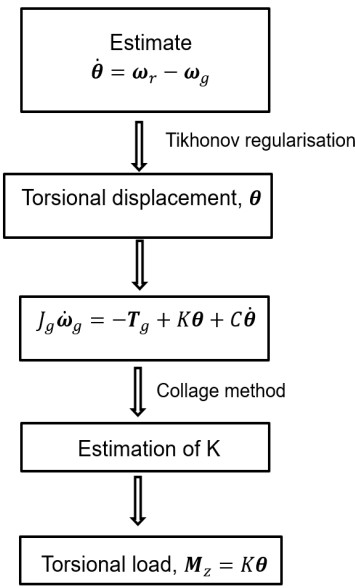

**Figure 2.** Flowchart depicting the inverse problem algorithm.

## 2.1 Tikhonov Regularisation

Given $\omega_r$ and $\omega_g$, $\dot{\boldsymbol{\theta}}$ is obtained by using Eq. (3) and then by numerically integrating $\dot{\boldsymbol{\theta}}$, the shaft torsional displacement ($\boldsymbol{\theta}$) is obtained. The numerical integration schemes require initial conditions as they march on time. However, the initial conditions are unavailable or inaccurate in practice. The lack of initial conditions (assumed usually 0) on the displacement are inconsistent with the real values that result in the phenomenon of baseline shift or drift which causes the position error to grow with time during the integration (Pahn et al., 2017). The effect of the lack of initial conditions on the reconstructed displacement obtained

using the time integration scheme is shown in Fig. 3. As seen in the figure, the numerical error is multiplicatively increased with time which results in a drift in the reconstructed displacement. As mentioned in the introduction, to minimize the numerical error due to the lack of initial conditions and to minimize the effect of measurement noise, a widely used regularization technique called Tikhonov regularization (Tikhonov, 1963) is employed here.

   Implementation of Tikhonov regularization on the velocity to obtain the displacement is not readily available in the literature

and hence the same is presented in the Appendix A.1 for the sake of completeness. By following the procedure outlined in Appendix A.1, the regularised torsional displacement ($\boldsymbol{\theta}$) is obtained as,

$$\boldsymbol{\theta} = \left(\frac{\mathbf{L}^2}{4} + \lambda^2 \mathbf{I}\right)^{-1} \frac{\mathbf{L}\mathbf{L}_a \dot{\boldsymbol{\theta}} \Delta t}{2}. \tag{4}$$

Here, the matrices $\mathbf{L}, \mathbf{L}_a$ and $\mathbf{I}$ are defined in the Appendix A.1, $\lambda$ is the regualrisation parameter, $\Delta t$ is the time interval and $\dot{\boldsymbol{\theta}}$ is the torsional velocity obtained from Eq. (3). The obtained regularised torsional displacement is compared with the actual

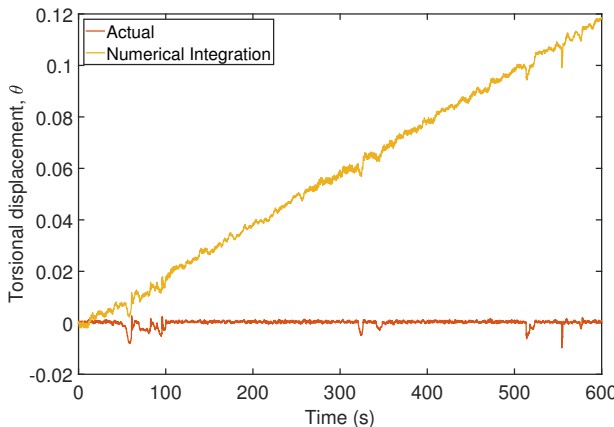

**Figure 3.** Comparison of time integration displacement with actual displacement.

displacement in Fig. 4 along with the numerical integration result. As seen in Fig. 4, there is a close match between the result
by Tikhonov regularisation and the actual displacement as compared to the numerical integration result.

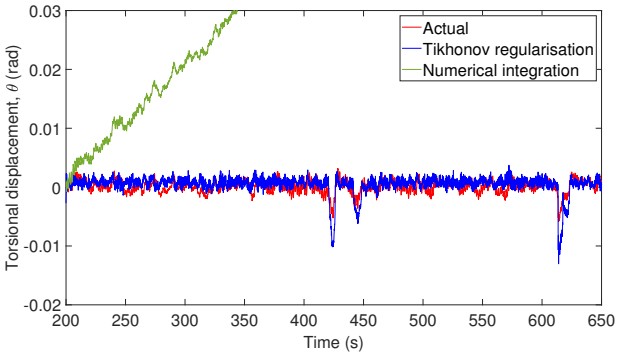

**Figure 4.** Comparison of the Tikhonov and time integration displacements with actual displacement.

## 2.2    Collage method

Upon estimating $\theta$ using Tikhonov regularisation, next step is to estimate the modal parameters using the Collage method (Kunze
and Vrscay, 1999; Groetsch, 1993) - a model-based approach. The mathematical formulation of the Collage method and its
implementation is given in Appendix A.2. The rotor equation (Eq. 1) cannot be used for the parameter estimation as there is
no information about the rotor torque in the SCADA measurement. Instead, the Collage method is employed on the generator
equation (Eq. 2) for estimating the modal parameters as the generator torque ($T_g$) can be readily obtained from the SCADA
data as, $T_g = P/\omega_g$. Accordingly, for the target function $\omega_g(t), t \in [t_0, t_n]$, the squared $\mathcal{L}^2$ collage distance (refer Appendix

| Wind turbine | Design $K$ ($N.m/rad$) | Estimated $K$ ($N.m/rad$) | % error |
|---|---|---|---|
| DTU-10 MW | 2.317E09 | 2.1785E09 | 5.98 |
| Vestas V52 | - | - | 6.98 |

**Table 1.** Comparison of estimated $K$ with design $K$ for a turbine variant where only the main shaft is flexible.

A.2) for Eq. (2) becomes,

$$\Delta^2 = \int\limits_{t_0}^{t_n} \left[ J_g[\boldsymbol{\omega}_g - \boldsymbol{\omega}_{g_0}] + \int\limits_{t_0}^{t_n} \boldsymbol{T}_g \, \mathrm{d}t - K \int\limits_{t_0}^{t_n} \boldsymbol{\theta} \, \mathrm{d}t - C \int\limits_{t_0}^{t_n} \dot{\boldsymbol{\theta}} \, \mathrm{d}t \right]^2 \mathrm{d}t, \tag{5}$$

where, $\boldsymbol{\omega}_{g_0}$ is the generator speed at time, $t = t_0$. Modal parameters will be obtained by minimising Eq. (5) with respect to $J_g, C$ and $K$. Upon obtaining $\boldsymbol{\theta}$ using the Tikhonov regularisation and $K$ using the Collage method, the torsional load is obtained as $\boldsymbol{M}_z = K\boldsymbol{\theta}$.

### 2.2.1 Application of the Collage method

To test the applicability and efficiency of the Collage method for the wind turbine drivetrain system, a verification study is undertaken by comparing the main shaft torsional stiffness obtained using the Collage method with its design value for two different wind turbines. This is done for the following two wind turbines, (i) DTU 10 MW (Bak et al., 2013) and (ii) Vestas-V52 (Vestas). For facilitating a comparison of the main shaft torsional response alone, the rigid variant of the turbine is chosen and this implies that the rotor and tower are rigid and the main shaft alone is considered to be flexible. By performing aeroelastic simulations on these turbines, the shaft torsional displacement $\boldsymbol{\theta}$ is obtained. Throughout the article, the aeroelastic simulation is performed in the DTU in-house tool called HAWC2 (Larsen and Hansen, 2007). HAWC2 (Horizontal Axis Wind turbine simulation Code 2nd generation) is used for calculating the wind turbine aeroelastic loads and responses in time domain. It uses multibody formulation to model the structure, blade element momentum (BEM) theory-based models for modelling the wind effects and hydrodynamic models for modeling the hydro-effects (in the case of offshore turbines) on the structure. Control of the turbine is performed through the dynamic link libraries (DLLs). Using the main shaft torsional load time series obtained from HAWC2 and, by minimising Eq. (5) concerning the modal parameters, $K, C$, and $J_g$ are obtained. Since for the estimation of shaft torsional load, only $K$ is needed among all the modal parameters, hence the same is compared with the design values. The estimated shaft stiffness and the stiffness from the aeroelastic model of the DTU 10-MW turbine along with percentage error are tabulated in Table. 1. Only the percentage error is given for the Vestas V52 turbine in Table. 1. As seen in the table, the estimated torsional stiffness values match well with the design values. If the torsional displacement ($\boldsymbol{\theta}$) is known, then the determination of torsional stiffness ($K$) from Eq. (5) is readily feasible as explained. However in practice, the shaft torsional displacement is unknown, and therefore the Collage equations may not be directly used to determine the shaft stiffness. In the following, the entire proposed methodology will be verified with the aeroelastic simulation results of the Vestas V52 turbine.

## 3 Verification of the method on V52 turbine simulations

Aeroelastic simulations are performed for the Vestas V52 turbine corresponding to the design load case (DLC 1.2) (IEC 61400-1) in HAWC2. The DLC 1.2 was run with eighteen 10-minute load simulations (three yaw directions: 0 deg. $\pm$ 10 deg and six turbulent wind seeds ) for each mean wind speed ranging from 4 $m/s$ to 26 $m/s$ in the interval of 2 $m/s$, which results in total 216 simulations. Each simulation uses 10-minute normal wind turbulence inflow with a sampling frequency of 50 Hz. From these 216 simulations, the inputs ($\boldsymbol{\omega}_r, \boldsymbol{\omega}_g$, and $\boldsymbol{T}_g$) for the proposed methods are obtained and by following the procedure

depicted in Fig. 2, the torsional loads are obtained and the same is compared with the simulation results. From the simulated $\boldsymbol{\omega}_r$ and $\boldsymbol{\omega}_g$, the torsional velocity $\dot{\boldsymbol{\theta}}$ is obtained using Eq. (3) and then the corresponding $\boldsymbol{\theta}$ for each simulation is obtained by using the Tikhonov regularisation (Eq. 4). By assuming that the same level of numerical noise (noise due to lack of initial conditions) presents in all the simulations of a particular mean wind speed, the optimal $\lambda$ parameter for regularization (refer Appendix A. 1 ) is estimated only once per mean wind speed. The estimated regularisation parameter ($\lambda$) for each mean wind

speed is presented in Fig. 5. As seen in the figure, a higher value of $\lambda$ at higher wind speeds indicates that more numerical damping is needed to suppress the numerical noise.

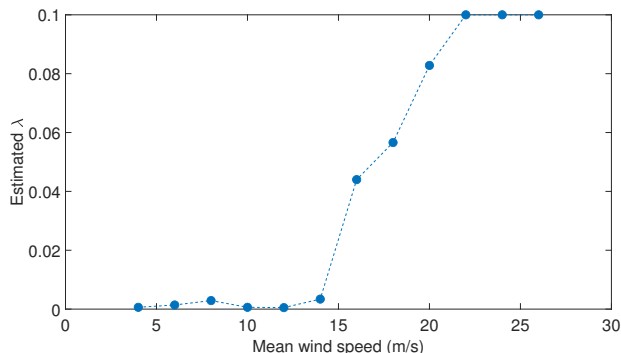

**Figure 5.** Estimated $\lambda$ for each mean wind speed.

Since the displacement is reconstructed from the velocity- a dynamic quantity, the reconstructed displacement will have zero mean and this displacement component is referred to as a dynamic component of the displacement ($\boldsymbol{\theta}_{dyn}$). However, the torsional displacement will have a contribution from the static load and the displacement due to the static load is referred

to as a static component of the displacement ($\boldsymbol{\theta}_{stat}$) (*i.e.*, the mean value of the displacement). The dynamic component of the displacement oscillates about the static component of the displacement. The regularized $\boldsymbol{\theta}_{dyn}$ for two representative mean wind speeds are compared with the dynamic component of simulated torsional displacement in Fig. 6. By removing the static displacement from the simulated displacement, the resulting dynamic component can be compared with the regularised $\boldsymbol{\theta}_{dyn}$ as shown in Fig. 6. As seen in the figure, the regularised $\boldsymbol{\theta}_{dyn}$ matches well with the simulated dynamic displacement for most

of the time except for the peak amplitudes.

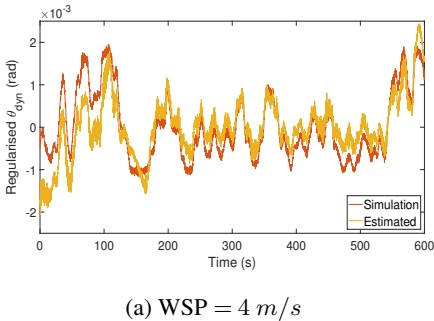

(a) WSP $= 4 \ m/s$

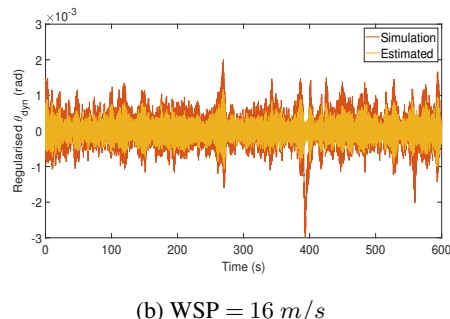

(b) WSP $= 16 \ m/s$

**Figure 6.** Comparison regularized $\boldsymbol{\theta}_{dyn}$ with the results from aeroelastic simulation.

Upon ensuring the correctness of $\boldsymbol{\theta}_{dyn}$, Collage method can be employed for $K$ estimation using the results of all the 216 simulations. Since $\boldsymbol{\theta}_{dyn}$ only available, Eq. (5) is modified as to have the dynamic components alone for the estimation of $K$ as,

$$\Delta^2 = \int\limits_{t_0}^{t_n} \left[ J_g[\boldsymbol{\omega}_g - \boldsymbol{\omega}_{g_0}] + \int\limits_{t_0}^{t_n} (\boldsymbol{T}_g - \boldsymbol{T}_{g_{mean}}) \, \mathrm{d}t - K \int\limits_{t_0}^{t_n} \boldsymbol{\theta}_{dyn} \mathrm{d}t - C \int\limits_{t_0}^{t_n} \dot{\boldsymbol{\theta}} \mathrm{d}t \right]^2 \mathrm{d}t. \tag{6}$$

By minimising Eq. (6), $K$ will be obtained for all the 216 simulations. Owing to uncertainty associated with $\boldsymbol{\theta}_{dyn}$ estimation along with the Collage method, three skewed distributions for $K$ can be determined at different mean wind speeds. For these distributions, the mode is a more stable estimate of the central tendency as it is least biased by outliers (Hedges and Shah, 2003). The mean of modes of the resulting probability density functions (pdfs) provides the resultant estimate of $K$. The relative error between the estimated $K$ value obtained by taking the mean of modes and the design value of the turbine is 12.06 %. At this

point, it is important to remember that since the inputs are from the HAWC2 aeroelastic simulation which does not account for gearbox dynamics, the estimated stiffness value has the contributions from the rotor and the main shaft only. As a result, the estimated $K$ is the resultant torsional stiffness of the main shaft including the blade edgewise stiffness contribution.

Even though the dynamic component of the displacement is sufficient enough for $K$ estimation as explained above, the mean value (i.e., the static component) of the displacement has a significant effect on the fatigue damage (Veldkamp, 2006). Hence,

it is important to estimate the same and this is obtained by solving the static problem of the drivetrain. By considering the static equilibrium of Eq. (1) with all parameters expressed on the low-speed side as follows,

$$K\boldsymbol{\theta}_{stat} = \boldsymbol{T}_{r_{mean}} = \frac{\boldsymbol{T}_{g_{mean}}}{\eta_{gen}}. \tag{7}$$

Here, $\eta_{gen}$ is the generator efficiency, which is 94.4 % for the V52 turbine. As an alternative way, one can use the overall efficiency of the wind turbine as well for the sake of completeness. However, the use of different efficiencies will not significantly

affect the outcome. From Eq. (7), $\boldsymbol{\theta}_{stat}$ is obtained as,

$$\boldsymbol{\theta}_{stat} = \frac{\boldsymbol{T}_{g_{mean}}}{K\eta_{gen}}. \tag{8}$$

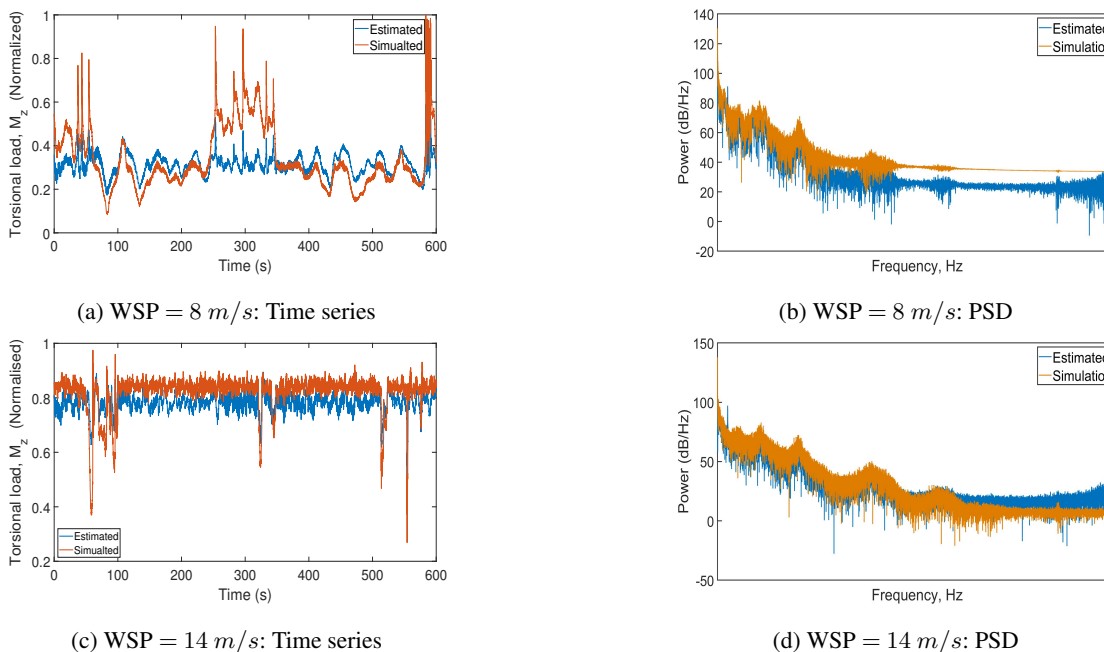

(a) WSP $= 8\ m/s$: Time series

(b) WSP $= 8\ m/s$: PSD

(c) WSP $= 14\ m/s$: Time series

(d) WSP $= 14\ m/s$: PSD

**Figure 7.** Comparison reconstructed time series and power spectral density (PSD) of the torsional load for two different mean wind speeds.

Finally, the static and dynamic components are superimposed to get the actual displacement ($\boldsymbol{\theta}$). This actual displacement will then be used to estimate the site-specific main shaft torsional load as shown by Eq. (9).

$$M_z = K\boldsymbol{\theta} = K(\boldsymbol{\theta}_{stat} + \boldsymbol{\theta}_{dyn}), \tag{9}$$

At this stage, the estimated torsional loads can be compared with the simulated torsional loads. The comparison of the torsional loads for two representative mean wind speeds are shown in Fig. (7). As seen in Fig. (7), all the important aspects of the time-series variation and the dominant frequency dynamics (low-frequency components - up to first three peaks ) are captured quite well in the estimated torsional load. The computational time for identifying the regularisation parameter for each mean wind speed is about 40 minutes in real-time and the stiffness estimation for the 10-minute simulation takes 14 s in real-time. The
above computations are performed on a single node of the high-performance computing cluster of DTU. The cluster has 320 nodes in total, each node consists of two Intel Xeon E5-2680v2 processors, and each processor consists of 10 cores running at 2.8 GHz. If the wind farm is of same type of turbine, then it is sufficient to estimate the stiffness for one of the turbines. However, the shaft torsional displacement needs to be estimated for all the turbines.

Upon estimating the torsional load, the torsional damage equivalent load (DEL) at each mean wind speed is calculated using
the following equation:

$$\mathrm{DEL} = \left( \frac{1}{N_{ref}} \sum_{i=1}^{N_{sim}} \left( \frac{1}{N_{sim}} \right) \sum_{k=1}^{k_n} \frac{N_{i,k} S_{i,k}^m(0)}{T_{sim}} \right)^{\frac{1}{m}}, \tag{10}$$

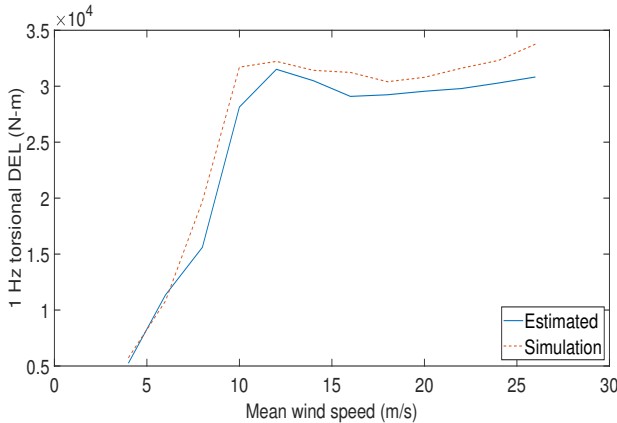

**Figure 8.** Comparison of the predicted DEL with the DEL computed from aeroelastic simulations over all mean wind speeds.

where, $T_{sim}$ is the duration of the load case, $N_{sim}$ is the number of simulations at each mean wind speed, $k_n$ is the total number of load cycles in a given time series, $N_{i,k}$ are the number of cycles at load amplitude $S_{i,k}(\sigma)$ determined with rain flow counting and $m$ is the Wöhler exponent. The zero mean load amplitude is obtained as (Veldkamp, 2006), $S_{i,k}(0) = S_{i,k}(\sigma) + MS_m$, where, $S_m$ is the mean load and $M$ is the mean stress sensitivity. The turbine shaft is made up of cast iron and Veldkamp (2006) has reported that for such material, the mean stress correction factor is $M = 0.19$. The Wöhler exponent for the cast iron of $m = 6$ (Veldkamp, 2006) is used in the present study. When $N_{ref}$ in Eq. (10) equals the component's lifetime in seconds, then the DEL has the frequency of 1 Hz. The computed torsional DEL using Eq. (10) for DLC 1.2 is compared with the simulated DEL and is shown in Fig. (8). For most of the mean wind speeds, the estimated DEL is in good agreement with the simulated DEL and the absolute error between these two at each mean wind speed ranges from $4\%$ to $12\,\%$. Higher error at higher mean wind speeds is due to the fact that the peak amplitudes in $\boldsymbol{\theta}_{dyn}$ are not captured well using the Tikhonov regularisation. The fact that the peak amplitudes are not captured in the reconstructed torsional displacements is typical for Tikhonov regularisation as there is a slight mismatch in the frequency spectra between the actual and reconstructed torsional oscillations (Lee et al., 2010). Another reason could be due to the fact that the pitch angle influences the main shaft torsional oscillation through the rotor torque and the rotor speed. However, the controller is maintaining a constant rotational speed beyond the rated wind speed, hence the instantaneous changes in the pitch angle are not propagated through the rotor speed to the main shaft oscillations. All these effects, in addition to the uncertainty in $K$ estimation, may resulted in a maximum error of $12\,\%$ in the estimated DEL.

## 4 Application on V52 turbine measurements

Upon verifying the proposed method, the drivetrain main shaft torsional loads are estimated from the SCADA measurements without any need for the aeroelastic model. SCADA measurements taken during January 2019 for the Vestas V52-850 kW research turbine installed at the DTU Risø site is utilized for this purpose. The measurement data consists of 4459 ten minute recorded cases with 50 Hz sampling frequency. Generator torque is used as one of the SCADA signal instead of the generator

| | |
|---|---|
| Minimum rotor speed | 1.7 $rad/s$ |
| Minimum power | 100 W |
| Minimum wind speed | 4 $m/s$ |
| Wind direction | [280°-320°] |

**Table 2.** Normal operation filter conditions.

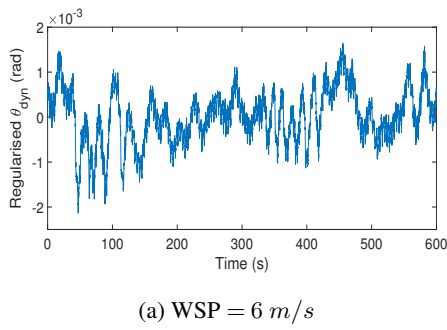

(a) WSP $= 6\ m/s$

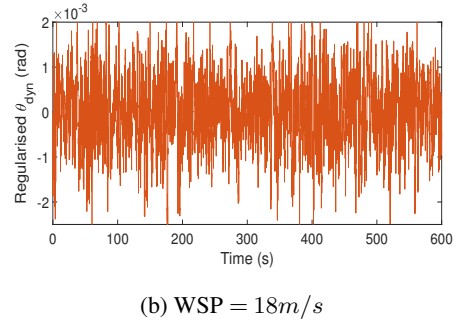

(b) WSP $= 18 m/s$

**Figure 9.** Regularised $\boldsymbol{\theta}_{dyn}$ for two mean wind speeds.

speed for this part of the study and the generator speed is obtained from the generator power and generator torque (on the low-speed side) SCADA signals as $\boldsymbol{\omega}_g = P/\boldsymbol{T}_g$ . Further, the measurement data corresponds to normal operation is filtered based
on the conditions given in Table 2 that results in 627 cases. It is important to note that based on the site sector assessment, the V52 turbine is in wake free condition between 280° and 320° wind directions.

Using the rotor speed and generator speed, $\boldsymbol{\theta}_{dyn}$ is calculated using the Tikhonov regularisation method and the obtained $\boldsymbol{\theta}_{dyn}$ for two representative mean wind speeds are shown in Fig. 9.

The regularisation parameter is obtained for each mean wind speed measurement using the L-curve criterion and the obtained
values not presented here for the sake of brevity. Subsequently, by applying the Collage method on Eq. (6) for all the 627 cases, the $K$ values are estimated for each load case and then the resultant $K$ value is obtained by taking the mean of modes of the resulting pdf as described in the previous section. At this point, it is important to remember that the inputs are from measurement, hence the estimated $K$ is a collective stiffness that has a contribution from all the drivetrain components including the gearbox. With the estimated $K$ value, $\boldsymbol{\theta}_{stat}$ is calculated for each load case using Eq. (8) and then the torsional load is
obtained from Eq. (9). The calculated torsional loads for two representative wind speeds are shown in Figs. (10a, 10b). After estimating the torsional loads for all the 627 cases, the identified loads are grouped according to the mean wind speeds range from 6 $m/s$ to 22 $m/s$ which are subdivided into 9 wind speed bins of 2 $m/s$ width each. Subsequently, the torsional DEL is calculated for each mean wind speed using Eq. (10) and the same is shown in Fig. (11). It is important to note that the DEL given in Fig. 8 (simulation) being the design load and the DEL given in Fig. 11 being the site-specific load, the difference
between these two can give an estimate about the remaining torsional fatigue life of the main shaft under normal operating

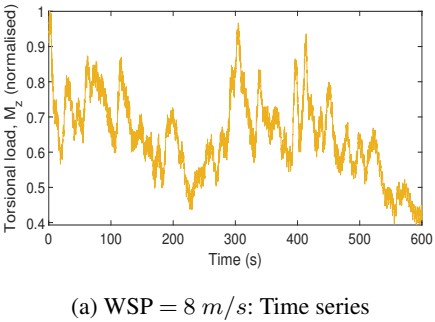

(a) WSP $= 8\ m/s$: Time series

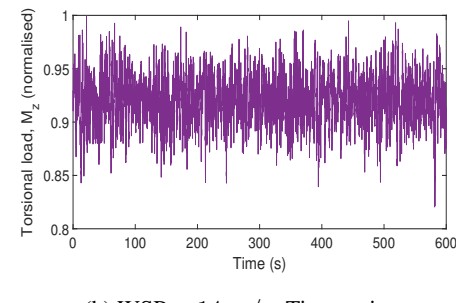

(b) WSP $= 14\ m/s$: Time series

**Figure 10.** Identified torsional loads ($M_z$) for two mean wind speeds.

conditions (Ziegler et al., 2018). Further, the estimated main shaft torsional DEL may be used to quantify the RUL of the drivetrain (Pagitsch et al., 2020). However, this is beyond the scope of the current work. It is also feasible to reconstruct the torsional load time series, which may be used an input to gearbox design tools to predict the loading within the gearbox.

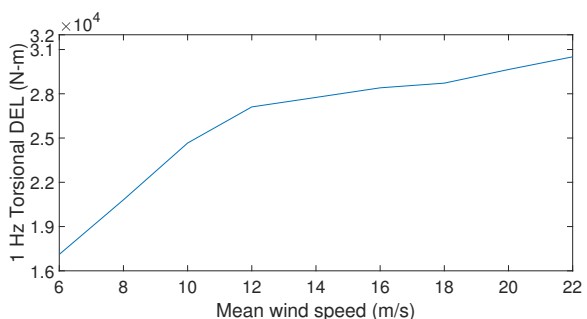

**Figure 11.** Estimated torsional DEL from SCADA measurements of Vestas V52 turbine.

## 5  Summary

A novel inverse problem-based approach is developed for estimating the main shaft torsional displacement and stiffness by using the high-frequency SCADA measurements. A mathematical model describing the coupled shaft torsional dynamics is used for this purpose. Numerical errors and the effect of measurement noise on the torsional displacement reconstruction are minimized through the Tikhonov regularization technique. Subsequently, the Collage method is used to estimate the main shaft coupled torsional stiffness. The estimated main shaft quantities are then used to identify the main shaft site-specific torsional

load. The proposed formulation is successfully verified for the main shaft torsional loads with the aeroelastic simulation of the Vestas V52 turbine. Upon verification, the methodology is extended to identify the site-specific main shaft torsional loads of the same turbine by using SCADA measurements. For this purpose, the measurement data from the DTU Risø site is utilized and the measurement data is filtered and calibrated for the turbine normal operation. Using the identified torsional loads,

the torsional DEL is obtained. Depending on the prior information about the stiffness value, one can either use the entire proposed methodology or follow the torsional displacement estimation part of the proposed methodology for the torsional load identification. Since the site-specific SCADA measurements are used in the analysis, the obtained loads can give a better estimate of the remaining fatigue life of drivetrain components. Monitoring the estimated loads can help in inspection planning and scheduling maintenance activities. As the proposed methodology does not require any design basis or an aeroelastic design basis, it can be used for wind turbines that possess high-frequency SCADA measurements for the estimation of the main shaft torsional load and DEL.

## Appendix

### A.1 Displacement Reconstruction using the Tikhonov Regularisation

By definition, the velocity $\dot{\boldsymbol{\theta}}$ is expressed as,

$$\dot{\boldsymbol{\theta}}(t) = \frac{\mathrm{d}\boldsymbol{\theta}}{\mathrm{d}t} \approx \dot{\tilde{\boldsymbol{\theta}}}(t), \tag{11}$$

where, $\dot{\tilde{\boldsymbol{\theta}}}(t)$ is the velocity obtained from Eq. (3) which can be considered as a measured velocity. As explained in Section 2.1, the lack of initial conditions in addition to the measurement noise leads to erroneous displacement. In order to minimise the error between the actual and measured velocities in the least square sense, following minimisation problem has to be solved,

$$\text{Min } \Pi_E(\boldsymbol{\theta}) = \frac{1}{2} \int_{t_0}^{t_n} (\dot{\boldsymbol{\theta}}(t) - \dot{\tilde{\boldsymbol{\theta}}}(t))^2 \, \mathrm{d}t. \tag{12}$$

Here, $\dot{\boldsymbol{\theta}}$ is the calculated velocity. In other words, Eq. (12) gives a measure of how well the actual velocity approximates the
measured velocity. By means of the trapezoidal rule, Eq. (12) is discretised as follows (Hong et al., 2008),

$$\Pi_E(\boldsymbol{\theta}) \approx \|\mathbf{L}_a(\dot{\boldsymbol{\theta}} - \dot{\tilde{\boldsymbol{\theta}}})\|_2^2 \Delta t, \tag{13}$$

where $\Delta t$ is the time interval of the discretization and $\mathbf{L}_a$ is the diagonal weighing matrix of order $(N+1)$ as,

$$\mathbf{L}_a = \begin{bmatrix} 1/\sqrt{2} & & & & \\ & 1 & & & \\ & & \ddots & & \\ & & & 1 & \\ & & & & 1/\sqrt{2} \end{bmatrix}. \tag{14}$$

Here, $N$ is the number of data points in $\boldsymbol{\theta}$ and for a 10-minute simulation with 50 Hz sampling frequency $N$ becomes 30000.
Further, the calculated velocity $\dot{\boldsymbol{\theta}}$ is discretised by the central difference rule and written in matrix form as,

$$\frac{1}{\Delta t} \mathbf{L}_c \boldsymbol{\theta} = \dot{\boldsymbol{\theta}}, \tag{15}$$

where, the central difference matrix $\mathbf{L}_c$ of size $(N+1) \times (N+3)$ and the displacement vector $\boldsymbol{\theta}$ of size $(N+3)$ are given as,

$$\mathbf{L}_c = \begin{bmatrix} 1 & 0 & 1 & \ldots & & \\ & -1 & 0 & 1 & & \\ & & \ddots & & & \\ & & -1 & 0 & 1 & \\ & \ldots & & -1 & 0 & 1 \end{bmatrix}, \qquad \boldsymbol{\theta} = \begin{pmatrix} \boldsymbol{\theta}_{-1} \\ \boldsymbol{\theta}_0 \\ \vdots \\ \boldsymbol{\theta}_n \\ \boldsymbol{\theta}_{n+1} \end{pmatrix}. \tag{16}$$

In the finite difference discretization of Eq. (16), some defined nodes are located outside of the domain considered (*i.e.*, domain here is time, $t \in [t_0, t_n]$ satisfying $t_0 \leq t \leq t_n$). These nodes defined by time steps $i = -1$ and $i = n+1$ are fictitious. These nodes are the dummy nodes that are used in solving the differential equations by the finite difference method (Lapidus and Pinder, 2011). Substitution of Eq. (15) into Eq. (13) leads to,

$$\text{Min } \Pi_E(\boldsymbol{\theta}) \approx \frac{1}{2}\left\|\frac{1}{2\Delta t}\mathbf{L}_a\mathbf{L}_c\boldsymbol{\theta} - \mathbf{L}_a\dot{\boldsymbol{\theta}}\right\|_2^2 \Delta t = \frac{1}{2}\left\|\frac{1}{2}\mathbf{L}\boldsymbol{\theta} - \mathbf{L}_a\dot{\boldsymbol{\theta}}\Delta t\right\|_2^2 \frac{1}{\Delta t}, \tag{17}$$

where, $\mathbf{L} = \mathbf{L}_a\mathbf{L}_c$. This minimisation problem is regularised for solution boundedness with a parameter $\lambda$, and given as,

$$\text{Min } \Pi_E(\boldsymbol{\theta}) \approx \frac{1}{2}\left\|\frac{1}{2}\mathbf{L}\boldsymbol{\theta} - \mathbf{L}_a\dot{\boldsymbol{\theta}}\Delta t\right\|_2^2 + \frac{\lambda^2}{2}\|\boldsymbol{\theta}\|_2^2. \tag{18}$$

The above minimisation problem is known as the Tikhonov regularisation and $\lambda$ is referred to as the regularisation parameter. Minimising Eq. (18) as,

$$\frac{\mathrm{d}\Pi_E}{\mathrm{d}\boldsymbol{\theta}} = \frac{1}{2}\left(\frac{\mathbf{L}^2\boldsymbol{\theta}}{2} - \mathbf{L}\mathbf{L}_a\dot{\boldsymbol{\theta}}\Delta t\right) + \lambda^2\boldsymbol{\theta} = 0, \tag{19}$$

yields the following quadratic equation in $\boldsymbol{\theta}$,

$$\boldsymbol{\theta} = \left(\frac{\mathbf{L}^2}{4} + \lambda^2\mathbf{I}\right)^{-1}\frac{\mathbf{L}\mathbf{L}_a\dot{\boldsymbol{\theta}}\Delta t}{2}, \tag{20}$$

where, $\mathbf{I}$ is the identity matrix of order $(N+3)$.

The choice of regularisation parameter $(\lambda)$ plays a crucial role in getting an optimal fit for the solution. Based on the knowledge about measurement errors, Hansen (2005) proposed two classes for the estimation of $\lambda$:

– methods based on knowledge of measurement errors

– methods that do not require details about measurement errors.

In the present scenario, the information regarding the measurement error is unknown, hence class two is used for the current study. In class two, there are three widely used methods (Nieminen et al., 2011): (i) quasi optimality criterion, (ii) Generalized gross validation (GCV), and (iii) L-curve method. Compared to the GCV method, the other two methods give a better estimate of $\lambda$ (Gao et al., 2016). Further, for larger problems, the quasi optimality method is computationally more expensive than the L-curve method. Owing to this fact, the L-curve method is used here for estimating $\lambda$. In L-curve method, the optimal $\lambda$ is the one which gives the maximum curvature in the L-curve between norm of the regularized solution $\alpha(\lambda) = \|\boldsymbol{\theta}_{reg}\|_2$ and norm of the residual $\beta(\lambda) = \left\|\frac{1}{2}\mathbf{L}\boldsymbol{\theta} - \mathbf{L}_a\dot{\boldsymbol{\theta}}\Delta t\right\|_2$ and the curvature of the L-curve is given by (Nieminen et al., 2011),

$$\kappa(\lambda) = \frac{\ddot{\alpha}\dot{\beta} - \ddot{\beta}\dot{\alpha}}{[\dot{\alpha}^2 + \dot{\beta}^2]^{3/2}}. \tag{21}$$

Substituting the optimal $\lambda$ obtained by finding the maximum curvature of Eq. (21) and $\dot{\boldsymbol{\theta}}(t)$ obtained from Eq. (3) in Eq. (20), the regularised torsional displacement $(\boldsymbol{\theta})$ is obtained.

## A.2 Modal Parameters Estimation using the Collage Method

For a given initial value problem (IVP),

$$x(t) = f(x,t), x(0) = x_0, \tag{22}$$

that admits a target solution $x(t)$, the associated Picard integral operator $\mathrm{T}$ is given by,

$$(\mathrm{T}u)(t) = x_0 + \int_{t_0}^{t_n} f(u(s),s)\mathrm{d}s. \tag{23}$$

The assumptions regarding the parameter estimation problem using the Collage method are listed as follow (Deng and Liao, 2009):

1. $x(t) \in [t_0, t_n]$ is a bounded solution; where, $t_0$ and $t_n$ are positive constants satisfying $t_0 < t_n$.

2. $f(u, x, t, \gamma_1, \cdots, \gamma_m)$ is continuous, where, $\gamma_i, i = 1, \cdots, m$ are the unknown modal parameters.

3. The exact solution $x(t)$ of the system (22) exists uniquely.

Here, the unique solution of the considered IVP is given by the fixed point $\bar{u}(t)$ of this Picard operator (Kunze et al., 2004). Accordingly, the Collage distance becomes, $(x - \mathrm{T}x)$ and then the optimal solution is the one which minimizes the squared $\mathcal{L}^2$ Collage distance (Here, $\mathcal{L}^2$ is 2-norm or square norm of a function). Also, unlike the conventional inverse problem which minimises the approximate error $\mathrm{d}(x - \bar{x})$, the Collage method minimizes the Collage distance $\mathrm{d}(x, \mathrm{T}x)$ which is an useful change as one cannot find $\bar{x}$ for a general $\mathrm{T}$ (Kunze et al., 2004). Further, the optimality of the Collage distance minimization is ensured as shown by Kunze et al. (2004). Accordingly, the $\mathcal{L}^2$ Collage distance has the form,

$$\Delta = \left( \int_{t_0}^{t_n} (x(t) - (\mathrm{T}x)(t))^2 \mathrm{d}t \right)^{\frac{1}{2}}. \tag{24}$$

Minimising the squared $\mathcal{L}^2$ Collage distance yields a stationarity condition, $\frac{\mathrm{d}\Delta^2}{\mathrm{d}\gamma_m} = 0$, that results in a set of simultaneous linear equations as a function of unknown modal parameters $(\gamma_m)$. The modal parameters are then obtained by solving those set of linear equations.

*Author contributions.* **W. Dheelibun Remigius**: Methodology, Formal analysis, Investigation, Validation, Writing - original draft. **Anand Natarajan**: Original idea, developing the scientific methods, Writing - review and editing, Supervision.

*Competing interests.* The authors declare that they have no known competing financial interests or personal relationships that could have appeared to influence the work reported in this paper.

*Acknowledgements.* This work has been funded by the Energy Technology Development and Demonstration Programme, Denmark (EUDP) [Grant No. 64017 -05114]. Investigations have been carried out at Department of Wind Energy, Technical University of Denmark.

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
