# Peer review of "Identification of wind turbine main shaft torsional loads from high-frequency SCADA measurements using an inverse problem approach"

_Wind Energy Science, 2021_

## Referee Comment (RC2)

**Referee Comments for wes-2021-25**

**Abstract**

The manuscript presents novel application of collage method with an integration technique in an inverse scheme to identify the simplified drivetrain model and its torsional deformation for identification of wind turbine main shaft loads. The collage method is used to identify system parameters, in which the torsional displacement is initially required. As such, an inverse method for the time integration based on Tikhonov regularisation is applied to retrieve the torsional displacement of drivetrain form the measured angular velocity of rotor and generator. The novelty and impact of this work is centred on the use of SCADA data only and that no knowledge on the drivetrain model is required. Sufficient amount of results using both simulated and actual measurement data have been presented. The manuscript generally investigates an interesting research question, which is appealing to both academic and applied research areas in wind turbine science and technology.

**General comments**

The developed methodology and the way it has been implemented within the paper are well explained. However, some key questions on the theoretical basics and practical applicability of the presented method may arise. As such, further clarification, including explanation of the limitation of the method or adjustment of the paper's arguments seem to be required.

The first question concerns the simplified mathematical model of the drivetrain (Eqs 1-3). This model is a compact two-disk representation, in which the inertia of the main shaft and that of the gear box and high-speed shaft are collected into the equivalent generator inertia. Therefore for the sake of consistency of the model, the stiffness term (K) seems to stand for the collective stiffness of the whole drivetrain rather than solely that of the main shaft [1]. However, the manuscript is taking whole drivetrain apart from the main shaft perfectly rigid, so above stiffness is assigned to the main shaft and will then be identified and utilised it to infer the main shaft loads, which may not be right for drivetrains with multi-stage gearbox and shafts. There are several references available (e.g. [2]) that explain the equivalent drivetrain models as the combination of springs in series. The angular displacement of such models are basically the summation of the individual displacements of the drivetrain components. A comparison with simulation results cannot simply validate the above assumption, as the turbine simulation tools are also usually including a simplified drivetrain model, as they aim to capture the global motion and loads of system.  In case that the contribution of the other components are known/found to be negligible, this has to be quantitatively demonstrated or at least be discussed adequately. It is not well described whether this work is identifying the main shaft load or the total equivalent torsional load of the drivetrain. Nevertheless, the manuscript's methodology and findings are still interesting. As such, it is recommended to either clarify this point or discuss the probable limitations.

The other point is regarding the manuscript's argument that it just needs SCADA data, which sounds compelling due to this data being readily available.  The suitable sampling rate of the SCADA data that is required for this methodology has not been discussed within the paper and needs to be closely clarified. This point becomes more important when the manuscript argues that their findings are beneficial to the calculations for life extension of wind turbine system. However, many of the existing turbines that would require life extension are equipped with SCADA systems with very low frequency output data, normally averaged values of order of minutes. It seems that this drawback is going to be removed by artificially increasing the SCADA signal sampling rate without providing any information

on the original and the resulted sampling rate. Therefore, it is a far question whether this methodology is really applicable to SCADA data or condition monitoring system's data is still required. As such, it is again recommended to either clarify this point or discuss the probable limitations.

Moreover, there are well-established integration techniques that are being used, even for online conversion of acceleration to velocity and displacement signals [3, 4]. In the absence of artificial noise in the simulated speed data, the type of drift shown in Figures 3 and 4 are apparently showing a linear trend due to the accumulation of the error from the initial value. This type of trend can be usually avoidable by the common digital filtering with restriction of frequency range within the pre-processing of the signal, particularly when the signal's mean value (static term) is going to be added later on separately, in which case the initial value of integration doesn't really matter. As such, the authors need to mention that the application of regularisation within a trapezoidal scheme is not the only available method to stably convert velocity into displacement.

**Minor comments:**

The derivatives for the inverse integration scheme is similarly given in Hong et al 2008, so please refer to this paper in the beginning of the corresponding section.

Paragraph 15: "and" missing after strain gauges.

Paragraph 75: This is questionable if older turbines really possess SCADA with a desirable sampling rate that suits the manuscript's methodology.

Paragraph 85: As discussed previously, the assumption of the fully rigid gearbox and higher stage shafts, particularly for a multi-stage systems, needs further clarifications.

Paragraph 190: computationally "more" expensive.

Paragraph 205: Please clarify those three yaw directions.

Paragraph 210: where is the noise coming from that needs to be damped. Please clarify.

Paragraph 220: "the" modes.

Paragraph 225: pds should be expanded.

Mixed use of symbols has to be avoided throughout the manuscript.

Eq 18, it seems that gearbox ratio is missing somewhere.

Fig 8, it seems that some constant or near constant offset exist (observable in both time and frequency plots), apparently theta_static is not appropriately identified. The plotted FFT is too noisy, one can use a better illustration, perhaps, a power spectrum with sufficient averaging to get more clear peaks, as at the moment it is barely possible to get a good picture of the main peaks.

Paragraph 20: what are the sampling rate details and how the SCADA data's frequency resolution was increased, is it practically legitimate? Please clarify.

The results using the actual data does not look to add substantial value to the manuscript, as it is not really supporting the validation of the methodology.

**Summary**

The manuscript deals with an interesting question, relatively well written and structured, and will be of interest to WES audience, both academic and industrial. At the moment, this methodology seems to give a first-pass clue on the drivetrain specifications, rather than being capable to identify the specific loads/parameters of main shaft. It was discussed above that further clarification on some fundamental assumptions and some less major aspects are required. The authors are encouraged to either amend these requested points or moderate their arguments. In summary, this work is intriguing and suitable for publication once the requested major revisions are addressed.

**References:**

[1] J. Berglind, R Wisniewski, and M Soltani, Fatigue load modeling and control for wind turbines based on hysteresis operators, in: 315 2015 American Control Conference (ACC), pp. 3721–3727, IEEE, 2015.
[2] P Girsang, J.S. Dhupia, E. Muljadi and M. Singh, L.Y. Pao, Gearbox and Drivetrain Models to Study Dynamic Effects of Modern Wind Turbines, NREL/CP-5500-58960
[3] A Brandta and R. Brincker, Integrating time signals in frequency domain – Comparison with time domain integration, Measurement 58 (2014) 511-519
[4] L Qihe, Integration of Vibration Acceleration Signal Based on LabVIEW, Journal of Physics: Conference Series1345 (2019) 042067, doi:10.1088/1742-6596/1345/4/042067

---

## Author Comment (AC1)

**Identification of wind turbine main shaft torsional loads from high-frequency SCADA measurements using an inverse problem approach**

The authors thank the reviewer for the valuable comments that improve the article quality. We have addressed all the reviewer's queries in detail in the following:

**Reviewer 1:**

**General comments**

**Structure, clarity and readability –** in its current form the paper is haphazard and difficult to follow. The introduction jumps between background information and descriptions of the methodology in the current work. The structure and flow of the paper overall is not clear, with the reader often left wondering where a given section is headed, or having to try and work out whether a certain concept is being newly introduced or if the author is referring to something already discussed. The paper needs to be reorganised such that introduction, background, methodology and results are all clear and distinct from one another – with the overall structure and flow of the paper made clearer from the start, and better signposted on the way through.

**Modelling and analysis presentation and associated discussions –** Descriptions of the modelling and analysis undertaken in this paper is currently lacking in clarity, completeness and context. The lumped parameter model is introduced without discussion of the approximations being made and the potential impacts this could have on model accuracy. Inputs to the model are not described in sufficient detail, e.g., frequency requirements for input SCADA data applied throughout this work, and the effect data frequency may have on the presented method, are not properly discussed. Note, when SCADA data is mentioned, one normally assumes 10 min mean values are being referred to, whereas high frequency SCADA (50Hz) is used here for at least part of the analysis. This is therefore a possible source of confusion for readers of the current paper. Also, the stiffness value being approximated is stated to contain contributions from components other than the main shaft, again the effect of this (good/bad/neutral) is not discussed, the fact is merely stated. The descriptions of the Collage method and Tikhonov regularisation, in their current form, are not appropriate. Much of the mathematics is simply stated without proper definitions for the various terms, or suitable descriptions to ensure the reader can follow what is being said. Notation also needs to be revised, with the same symbols currently representing both scalar and vector quantities. While I applaud a proper treatment of the mathematics, this is only useful if the formulations can be followed and reproduced by the reader. This requires careful definitions for all terms and better descriptions of how the resulting algorithms are implemented. I suggest that the authors consider providing a shorter description of each methodology, including why it is necessary and roughly how it works (e.g. provide an analogy), along with the resulting equation to be solved (e.g. Equation 6 for the collage method) and the solution method used. Then, the full

mathematical details can be referred to an appendix where they can be fully and carefully described, without derailing the flow of the overall paper.

**Context, motivations and claims** – While undoubtedly useful to the wind energy field, context regarding the wind industry is somewhat lacking from the paper in its current form. For example: is main-shaft fatigue currently of concern, are there available failure rates for this component, has it been much researched in the literature to date? As mentioned above this work seems to assume high frequency SCADA data for its analysis, but that isn't necessarily standard, especially for older turbines. The paper claims that the method is useful for older turbines, but what if they only have 10-min SCADA available? The paper shows certain levels of error for damage equivalent loads estimates, what does that accuracy level mean in terms of usefulness in the field or an ability to predict remaining useful life? Claims are made regarding remain useful life, but the link to that from results in the paper is non-trivial given variability of conditions and potentially incomplete histories of operating data – statistical analysis would likely be required. Note the above points don't necessarily all have to be bottomed out fully, but they should be discussed at some level in this work.

Response:

The authors have restructured the article by addressing the reviewers general and specific comments. By addressing the specific comments, the authors feel that the general comments will also be addressed. To address the reviewer's general and specific comments that concerns the introduction, the same has been re-written as follows (with the scope of each paragraph described in square parenthesis for explanatory purpose):

[Introduction-para 1]: Monitoring of wind turbines within wind farms is increasingly becoming very important due to the need to detect specific turbines that show anomalous behavior, plan inspections or preventive maintenance, and compute the remaining useful life of specific structures. Site-specific dynamic quantities such as structural response and modal parameters could assist in the condition monitoring of wind turbines.

[Background – para 2]: In general, the structural responses of the wind turbines are measured using accelerometers and strain gauges (Koukoura et al., 2015; Pahn et al., 2017; Norén-Cosgriff and Kaynia, 2021) and an output-based operational modal analysis (OMA) (Wang et al., 2016) technique is employed on the measured structural response for the identification of the modal parameters. A recent comprehensive review of different OMA methods can be found in Zahid et al. (2020). Upon identification of the modal parameters together with the measured structural response, an inverse problem-based technique is employed for the estimation of the site-specific loads (Pahn et al., 2017). Alternatively, one can use strain gauge-based load sensors to measure the site-specific loads directly (61400–13., 2016).

[Need for a new method in conjecture with the existing ones – para 3]: The addition of new instrumentation to existing turbines, such as the installation of strain gauges, accelerometers can be costly and require repetitive calibration and synchronization of their measurement signals with the turbine computer. Most wind farm operators also do not possess the aeroelastic design parameters of their wind turbines and hence one cannot simulate the mechanical loads acting on their wind turbines. Monitoring turbine primary structures through existing Supervisory Control and Data Acquisition (SCADA) system-based measurements can allow cost-effectiveness and provide valuable information to the wind farm operator. Usually, such monitoring through SCADA only provides information on power performance and not regarding turbine structural integrity. Since there are two SCADA signals (i.e., rotor speed and generator speed) related to the torsional oscillations of the main shaft of wind turbine drivetrains, the same can be used to quantify the torsional dynamics of the main shaft.

[Novelty and problem formulation – para 4]: For this purpose, an inverse problem-based approach is developed here to determine the dynamic quantities of the main shaft of a wind turbine, using existing high-frequency SCADA measurements such as the rotor speed and generator speed. This is a cost-effective alternative approach that is proposed for the main shaft to estimate both the torsional structural response and modal parameters directly from high-frequency SCADA measurements without using any additional measurement sensors or an aeroelastic design basis of the wind turbine. The proposed inverse problem approach is a model-based approach whereby a mathematical model concerning the shaft torsional dynamics will be utilized to obtain both the torsional displacement and the coupled torsional stiffness in a continuous time domain. It involves Tikhonov regularisation for regularising the measurement data and the collage method for estimating the torsional stiffness.

[Methodology – para 5]: The concerned mathematical model comprised of differential equations will be solved for the shaft torsional velocity with high-frequency rotor and generator speed measurements as inputs. Subsequently, the main shaft torsional displacement is obtained by numerically integrating the shaft torsional velocity. However, numerical integration is based on time-marching algorithms, and the lack of initial conditions makes displacement reconstruction an ill-posed problem (Hong et al., 2008). Further, these time marching algorithms are sensitive to measurement noise, and they even get amplified during the time marching procedure which results in inadmissible errors in the reconstructed displacement. This inaccurate displacement also leads to drastic errors in the system modal parameter estimation. Hence, one needs to go for regularisation techniques to smoothen the reconstructed displacement. Hansen (2005) discussed the nature of various ill-posed problems and presented several solution methodologies. Though there are many regularisation techniques available, Tikhonov, truncated singular value decomposition (SVD) and nuclear norm are a few of the popular techniques (Aarden, 2017). Among all these regularisation techniques, Tikhonov regularisation (Tikhonov, 1963) has been widely used in many engineering applications (Ronasi et al., 2011; Hào and Quyen, 2012; Bangji et al., 2017; Nieminen et al., 2011) and it has been studied extensively in the field of inverse problems

(Hansen, 2005). Further, digital filters and frequency domain integration approach (FDIA) are also widely used techniques in the literature to reconstruct displacements from measured accelerations (Brandta and Brincker, 2014, Qihe 2019 and Lee et al., 2010). However, digital filters such as impulse response filters (IIR) and finite response filters (FIR) have several drawbacks when reconstructing the low-frequency displacements as is the case here (Lee et al., 2010). On the other hand, the FIDA methods are sensitive to the time interval of the measurements (Lee et al., 2010). It is shown by Lee et al., that the Tikhonov regularisation is better suited for low-frequency dominant structures. Hence, the same has been employed in the present work. Tikhonov regularization minimizes the error using the least-square criterion and by means of numerical damping, it also minimizes the effect of the measurement noise.

[para -6 ] Upon obtaining the regularised shaft torsional displacement, the same mathematical model is utilized to obtain the shaft stiffness. For this purpose, the collage method - a model-based system identification technique - is used (Kunze and Vrscay, 1999; Groetsch, 1993). This method has been successfully applied for the system identification in various differential equations-based problems such as boundary value problems (Kunze et al., 2009), reaction-diffusion problems (Deng et al., 2008) and elliptic problems (Kunze and La Torre, 2016). The collage method is used to convert the inverse problem of system identification into a minimization problem of a function of several variables (for example unknown system parameters) and then the corresponding minimization problem is solved using a suitable algorithm. The minimization procedure is referred to as Collage coding and it is a greedy algorithm that seeks to construct an approximate solution to a target solution in one go. Hence, unlike other inverse problem methods, one need not solve the forward problem in an iterative manner (Deng and Liao, 2009). Further, the model-based collage method is simple, easy to implement and computationally inexpensive as compared to the output-based OMA methods.

[Application and limitation of the proposed work – para 7]: The estimated shaft torsional stiffness and displacement are further used to identify the site-specific shaft torsional load. This novel methodology can potentially benefit wind farm owners since both the property of the structure in terms of its stiffness and the structural response and the site-specific load can be determined without requiring additional sensors or information from the wind turbine manufacturer. The significance of estimating the main shaft torsional load is that it affects the fatigue performance of other drivetrain components such as gearbox and planetary bearings (Dong et al., 2012; Gallego-Calderon and Natarajan, 2015; Ding et al., 2018). Hence, the same site-specific torsional load may be used as an input for quantifying the remaining useful life (RUL) (Ziegler et al., 2018) of the main shaft, gearbox, and other drive train components as well. The estimation of RUL/ yearly damage does not require additional historical weather data and condition monitoring data as the wind speed and wind direction measurements are available in the SCADA measurements. However, this is beyond the scope of present work. Further, the proposed approach requires that the sampling frequency of the SCADA measurement be significantly higher than the dominant frequencies of the drivetrain torsional oscillations (i.e., 1p and 3p rotor excitation

frequencies and torsional natural frequencies). As a result, the proposed method cannot be used for the turbines that have measurements in terms of 10-min SCADA statistics.

**Specific comments**

1) *Page 2, lines 30-44* - why list all of these methods if you're not going to discuss them? Beyond stating that they exist its necessary to tell the reader something about them, e.g. pros, cons, what's relevant to the current study, why you used the method you did instead of these.

Response:

The concerned sentences that list various OMA methods have now been removed. Instead, studies that use such OMA methods for wind turbine system identification are listed with appropriate references that discuss their pros and cons. Hence, we have not repeated the pros and cons of these methods herein. These existing methods require the system response *a priori* for identifying the modal parameters. Usually, the system response is measured using accelerometers and strain gauges. The addition of new instrumentation to existing turbines, such as the installation of strain gauges, accelerometers can be costly and require repetitive calibration and synchronization of their measurement signals with the turbine computer. Hence, an alternative cost-effective approach is proposed here to estimate both the main shaft structural response and modal parameters directly from high-frequency SCADA measurements without using any additional measurement sensors. We have rewritten the concerned parts of the introduction. Please refer to para 2 and 3 of the rewritten introduction that is given as a response to the general comments.

2) Page 2, lines 42-43 – "However, to the best of the authors' knowledge, there is no such study available on estimating the structural response of a wind turbine component from SCADA measurements." You mentioned previously that many studies exist which consider OMA for wind turbines, do these not use SCADA data at all? Maybe describe them in more detail or list what they do use to make this claim easier to interpret.

Response:

The concerned sentences (i.e., L 42-43) have been removed from the revised article. However, as noted in the previous reply, the existing OMA methods use measured structural response for the identification of the modal parameters, whereas the proposed method uses high-frequency measurement to estimate both the structural response (torsional displacement) and the modal parameters (Shaft stiffness).

*3)* Page 2, lines 45-47 - We're now into methodology and haven't even reached the problem formulation yet. This is why splitting into intro, background, methodology seems to make sense. You can then give the summary overview of the types of models to be used in the intro, and then just go straight to the literature in background, that makes the flow feel better.

Page 3, line 60 - Again, this should be in Background - well in the summary of the methods at least, wherever that ends up being.

Response:

Estimating the main shaft torsional dynamic response and properties from high-frequency SCADA measurements is the scope of the work. This is done using an inverse problem-based approach that consists of Tikhonov regularisation for regularising the measurement data and the collage method for the system identification. As per the reviewer's suggestion, the introduction part concerning the methodology has been rewritten in such a way that the problem formulation is followed by an explanation of the methodology. Please refer to the re-written introduction.

*4)* Page 3, lines 73-74 – "Hence the same site-specific torsional load can also be used to quantify the RUL of gearbox and other drive train components as well". These are very big claims that need to be demonstrated if they are being made in the paper. Also in your drivetrain model don't you assume the gearbox is completely stiff? Does that not impact your ability to make conclusions regarding the gearbox? These are some of the reasons why I feel a better discussion of the applied model and its underlying assumptions is necessary.

Response:

Studies by Dong et al., 2012; Gallego-Calderon and Natarajan, 2015; Ding et al., 2018 showed that the torsional load of the main shaft can significantly affect the fatigue performance of the drivetrain components such as gearbox bearings. Hence, the estimated site-specific main shaft torsional loads using our proposed methodology may be used as an input for quantifying the RUL of the other drivetrain components. However, the specific quantification of the RUL of the gearbox is beyond the scope of the current work. The assumption regarding the drivetrain model will be revised as per the reviewers' suggestions. Accordingly, if the input data (i.e., high-frequency SCADA) accounts for the gearbox flexibility, then the estimated torsional load also has the contribution from the gearbox. This will be discussed in more detail while addressing the query about the modelling assumption.

We have rewritten the concerned paragraph in the manuscript. Please refer to the re-written introduction (para 7).

5) Page 3, lines 74-76 – Older turbines will likely not have a full service history of 1Hz SCADA data, how is the method applied in that case? Are you proposing that new data can be collected and used to estimate yearly damage for those turbines? That would also need some linking to historical weather records I'd imagine. This is not a trivial claim to make so more discussion is necessary.

Response:

We thank the reviewer for pointing out this fact. The proposed approach requires that the sampling frequency of the SCADA measurement be significantly higher than the dominant frequencies of the drivetrain torsional oscillations (i.e., 1p and 3p rotor excitation frequencies and torsional natural frequencies). As a result, the proposed method cannot be used for the turbines that have measurements in terms of 10-min SCADA statistics. The larger the turbine, the lower are the dominant frequencies of the torsional oscillations, hence for large size turbines the sampling frequency of SCADA measurements can be lower. In addition, we have used the phrase high-frequency SCADA in the article title as well. Also, the authors are not proposing any method for the new data collection. With the existing high-frequency SCADA measurements, the site-specific torsional load can be estimated. This can be used to estimate the yearly damage without historical weather records as the wind speed and wind direction measurements are available in the SCADA measurements.

We have rewritten the concerned paragraph in the manuscript. Please refer to the re-written introduction (para 7).

6) Page 3, line 86 – "gearbox is perfectly stiff". This approximation, among others, needs discussion and perhaps references, is it reasonable to assume this? What are the likely errors from it? Do any other papers deal with this type of approximation and show its effect?

Page 3, line 88 – "The main shaft is modelled by an inertia free viscously damped torsional spring". Same as for gearbox, the validity and effect of these approximations is necessary to allow the reader to understand how this model relates to the real world.

Page 4, line 98 – "the high frequency gearbox dynamics do not play a significant role in it." Do you have a reference for this claim? It needs to be demonstrated, referenced or at least discussed.

Page 4, line 99 – "Hence, the two-mass model is sufficient enough to model the shaft torsional dynamics for the wind turbine normal operations as it includes both the low frequency modes" This is a claim, can you back it up with evidence?

Response:

The two-mass model is widely used to study the torsional oscillations of the mechanical drivetrain of a wind turbine [1-5] because it captures the underlying free-free torsional frequency. This is the main driving frequency for the main shaft torsion. Though the governing equations (i.e., Eqs. (1-3) of the manuscript) remain the same for all the considered work. The underlying assumption regarding the stiffness of the various components in the drivetrain is different for each model and the definition for the 'K' in Eqs. (1-3) in each of these works is given in the following table:

| Model | Considered stiffnesses for K |
|---|---|
| Boukhezzar et al. [1] | Rotor, generator and the main shaft |
| Shin et al. [2] | Main shaft |
| Berglind et al. [3] | Collective stiffness with rigid gearbox |
| Novak et al., [4] | Main shaft |
| Singh & Santoso, [5] | Main shaft and high-speed shaft, gearbox dynamics dynamic neglected |
| Girsang et at., [6] | Stiffness of all components were considered. |

As seen in the table, the modelling assumptions regarding the gearbox and main shaft are inspired from [1-5]. However, for the inverse-problem based approach, the inputs to the drivetrain model (i.e., the rotor and generator speed) determine the contribution of each component to the 'K' value. For example, if the inputs are from the HAWC2 aeroelastic simulation then the estimated stiffness value has the contributions from the rotor and the main shaft only since the simulation does not account for the gearbox dynamics. If the inputs from the measurements are used, then the estimated stiffness is a collective stiffness that has a contribution from all the components.

To be consistent with all the inputs, the modelling assumptions adopted by Girsang et al.[6] has been referred in the revised manuscript (i.e., by considering the contributions from all the components in the drivetrain). Nevertheless, these modelling assumptions will not affect the methodology proposed herein.

Further, the main reason for ignoring the gearbox dynamics in some of the studies is that the torsional oscillation of the main shaft is mainly dominated by low-frequency components and hence the high-frequency gearbox dynamics will not significantly alter the main shaft torsional oscillations [4,5].

The concerned sentences (i.e., page 3, line 86 and line 88 and Page 4, line 98 and 99) will be removed from the revised manuscript. Accordingly, the discussion about the inputs to the model will be included along with the reference (Girsang et al., [6]) in the revised manuscript.

7) Page 4, line 100 – "Also, given the system parameters and rotor and generator torques, the two-mass model is capable of predicting the shaft torsional displacement as close as that of the full-fledged aeroelastic simulation as shown in Fig. 2." Another un-evidenced claim. Please discuss.

Response:

Since the modelling assumption is revised then the concerned figure does not add any value to the paper and hence it will be removed from the manuscript.

8) Page 4, line 102 – HAWC2 needs better description, what does this include (BEM model + multibody dynamics? etc.). Not all readers will be familiar with this code.

Response:

HAWC2 (Horizontal Axis Wind turbine simulation Code 2nd generation) is an aeroelastic simulation tool that is used for calculating the wind turbine loads and responses in time domain. It uses multibody formulation to model the structure, blade element momentum (BEM) theory-based models for modeling the wind effects and hydrodynamic models for modeling the hydro-effects (in the case of offshore turbines) on the structure. Control of the turbine is performed through the dynamic link libraries (DLLs).

The above description will be added to the revised manuscript in addition to the existing reference for the HAWC2.

9) Page 4, line 104 – "In forward problem approach" You are using lots of terminology that has not been defined, don't assume everyone knows what this means.

Response:

The concerned sentence will be modified as follows:

Given the modal parameters (Jr;Jg;C;K) and external torques (Tr and Tg), Eqs (1-3) are solved for omega_r; omega_g and theta, which is known as a forward problem approach [7].

10) Page 4, line 109 - How do you differentiate the theta values? Finite difference can go wrong for 'noisy measurements', do you apply a filter?

Response:

It is indeed true that the finite difference methods are highly sensitive to measurement noise. For this purpose, a regularisation technique called the Tikhonov regularisation was used in the study. It

minimizes the error due to numerical and measurement noise using the least-square criteria by means of numerical damping. The implementation of this regularisation technique is discussed in the latter part of the paper. Refer to Section 2.2 of the manuscript.

11) Page 5– see comments in 'General' concerning presentation of mathematical methods in the paper. Too high a burden us placed on the reader here, you need to decide what to keep, what to add and what to move to an Appendix. Currently readers would struggle to recreate these methods from the descriptions provided and the overall paper gets bogged down in technical details which are important but probably best in an Appendix and with improved definitions and descriptions.

Response:

Only a short description of the collage method and Tikhonov regularization will be presented in the methodology section. The detailed equations on the collage method and the Tikhonov regularization will be moved to the Appendix as per the reviewer's suggestion.

12) Page 6, Equation 6– is there an issue here? Should the 'u' in the integral operator now be 'theta'?

Response:

The authors thank the reviewer for pointing out the typo error. This will be corrected in the revised manuscript.

13) Page 6, line 137 – "hence it is important to check" This sentence does not necessarily hold. Just because something hasn't been tried it doesn't mean it should. There should be a better justification than that alone.

Response:

The concerned sentence will be modified as follows:

To test the applicability and efficiency of the collage method for the wind turbine drivetrain system, a verification study is undertaken by comparing the modal parameters obtained using the collage method with its design value for two different wind turbines.

14) Page 6, line 150– "As explained earlier," by this point the earlier descriptions have all merged with other details, another reason to have a clear, concise description of the methodology somewhere which is easily remembered and referred back to. Remember readers might not be familiar with these types of method so much will be new to them, some of these descriptions feel like they

assume quite a high level of familiarity. Also I'm not sure 'time integration' was specifically mentioned earlier in the paper.

Response:

The concerned sentence will be modified as follows:

Given omega_r and omega_g, theta_dot is obtained by using Eq. (3) and by numerically integrating the theta_dot, the shaft torsional displacement (theta) is obtained.

15) Page 7, Figure 3 – It is not clear to the reader why the numerically integrated estimate of theta diverges here. A 'lack of initial conditions plus measurements noise' is given as a reasons, but that doesn't help explain why. Is it because noise leads to greater 'areas under the curve' when integrating? But I'd expect that effect to give both higher and lower values. Please give a more detailed description, or analogy even, that gives the reader a sense for why this happens.

Response:

The lack of initial conditions (assumed usually 0) on the displacement are inconsistent with the real values that results in the phenomenon of baseline shift or drift which causes the position error to grow with time during the integration [8]. The sentence along with reference [8] will be added to the revised manuscript.

16) Page 7, Figure 3 –There seems to be a lack of clear definition for what theta is. I assumed it was displacement of the shaft, normally taken in-line with one of the blade roots. But that definition would see theta move from 0 to 2*pi repeatedly, which is not what is happening for the 'Actual' theta in Figure 3. Now it seems that theta is instead the torsional deflection in a moving frame which rotates with the shaft. The definition needs to be made clear when the model is introduced as otherwise this sort confusion can arise. Maybe add to Figure 1 to show theta more explicitly there?

Response:

The theta refers to the torsional displacement of the shaft. It is an instantaneous change of the shaft torsional displacement about the static equilibrium position in an inertial/fixed frame of reference. This will be added to the revised manuscript while introducing theta.

17) Page 7 and 8– Mathematical presentation again. Please see previous comments. Additionally, in Equation 8 it is not clear what we are minimising over, in Equation 9 theta now denotes a vector

(please use new notation when going from scalar to vector and make all definitions explicitly). These same developments include mention of 'fictitious nodes' with no description. Overall the reader is left struggling to understand what is happening and why, with little hope of being able to reproduce what is done here.

Response:

The formulation developed on pages 7 and 8 of the manuscript is used to regularize the influence of the numerical and measurement noise on the reconstruction of the torsional displacement. This is achieved by minimizing the error between the actual and measured velocities in the least square sense as in Eq. 8. The L2-norm of the residual is being minimized in Eq. (8). In other words, Eq. (8) gives a measure of how well the actual velocity approximates the measured velocity. By following the procedure outlined on pages 7 and 8, the concerned minimization problem given by Eq. (8) is reduced as a quadratic equation in Eq. (16). By using Eq. (16) with the measured velocity, one can obtain the reconstructed torsional displacement that is regularized for the numerical and measurement noise.

In the finite difference discretization of Eq. (11), some defined nodes are located outside of the domain considered (i.e., domain here is time, $t \in [0, n]$). These nodes defined by time steps $i=-1$ and $i= n+1$ are fictitious. These nodes are the dummy nodes that are used in solving the differential equations by the finite difference method [9]. This definition along with reference [9] will be given in the revised manuscript.

As per the reviewer's suggestion shorter descriptions of the collage method and Tikhonov regularisation along with the necessary equations and solution method will be given in the problem formulation. Detailed mathematical explanations will be shifted to the Appendix section. Further, all the mathematical symbols will be revisited in terms of their definition and notations in the revised manuscript.

18) Page 8, line 180 – "Since ten-minute SCADA measurements with a sampling frequency of 50 Hz are considered for theta(t) estimation, n becomes 30000." Now SCADA of much higher frequency is mentioned. This is even less common that 1s data. Assumptions and requirements concerning data from the wind turbine needs much clearer description and discussion throughout the paper. Also, if 50Hz is used here, will the method suffer is that's not available? Did you test to see what you can get away with at lower frequency?

Response:

The proposed approach requires that the sampling frequency of the SCADA measurement be significantly higher than the dominant frequencies of the drivetrain torsional oscillation. This explanation

will be given in the introduction as well as in the concerned place of the revised manuscript. (Please refer to the response of query 5).

19) Page 8, line 180 –Now thatn is becoming large it seems that a discussion of required computing power and computational times is necessary. Does this take a long time to run? How about if you were assessing a lot of data for remaining useful life analysis across a whole wind farm? I don't believe any such discussions are undertaken in the manuscript currently.

Response:

The computational time for identifying the regularisation parameter for each mean wind speed is about 40 mins in real-time and the stiffness estimation for the 10-min simulation takes 14 s in real-time. The above computations were performed on a single node of the high-performance computing cluster of DTU. The cluster has 320 nodes in total, each node consists of two Intel Xeon E5-2680v2 processors, and each processor consists of 10 cores running at 2.8 GHz.

If the wind farm of the same type of turbine, then it is sufficient to estimate the stiffness for one of the turbines. However, the shaft torsional displacement needs to be estimated for all the turbines.

These discussions will be included in the revised manuscript.

20) Page 9, line 209 – "the inputs". Again, what frequency is this data at?

Response:

The sampling frequency of the inputs is 50 Hz. The concerned sentence will be modified as follows:

The DLC 1.2 consists of 18 simulations (three yaw directions (i.e., 0 deg, ± 10 deg) and six turbulent wind seeds ) for each mean wind speed and the mean wind speed is ranging from 4 m/s to 26 m/s in the interval of 2 m/s, which results in a total of 216 simulations. Each simulation is of 10-min duration sampled at 50 Hz.

21) Page 10, Figure 5 – The flow chart is blurry and appears squashed. Please remake as a clearer vector image. Also I'd include this in a 'methodology' section earlier in the paper to help make the process clearer earlier in the paper.

Response:

A new flowchart of better quality will be included in the revised manuscript and as per the reviewer's suggestion, the same will be given in the ending of Section 2.

*22)* Page 10, line 216 – "At this point, it is important to realize that the static component of the displacement does not... and hence only the dynamic component of the torsional displacement" Neither of the 'static' or 'dynamic' component of torsional displacement is defined in the paper and this is the first mention that either of them get.

Response:

Since the displacement is reconstructed from the velocity- a dynamic quantity, the reconstructed displacement will have zero mean as shown in Fig. 7 and this displacement component is referred to as a dynamic component of the displacement (theta_dyn). However, the mean torsional displacement will have a contribution because of the static load and this contribution is referred to as a static component of the displacement (i.e., the mean value of the displacement). The dynamic component of the displacement oscillates about the static component of the displacement.

These definitions will be included into the revised manuscript.

*23)* Page 11, line 219 - "By removing the static displacement from the simulated displacement, the resulting dynamic component can be compared with...". Is that useful? Is static component important for damage in the shaft? Does is relate to the mean about which fluctuations are occurring? None of this is clear and so the highlighted line of text is also unclear regarding what is being compared and how the comparison is to be interpreted.

Response:

Since the reconstructed displacement is the dynamic component of the torsional displacement, the same is compared with the dynamic component of the displacement obtained from HAWC2 simulations in Fig. 7. Upon ensuring the correctness of the reconstructed displacement, the torsional stiffness of the drivetrain is estimated using the collaged method using the dynamic component of the displacement. However, the mean value (i.e., the static component) of the displacement has a significant effect on the fatigue damage [10]. Hence, it is important to estimate the static component of the shaft torsional displacement. This is obtained by solving the static problem of the drivetrain. Finally, the static and dynamic components are superimposed to get the actual displacement. This actual displacement will then be used to estimate the site-specific loads and fatigue damage.

The discussion will be included in the revised manuscript.

*24)* Page 11, line 228 – "12.06%". This sounds good but how good is it in the context of the problem at hand? How else might it be approximated and if there are other methods how does this compare?

Response:

The 12.06% deviation in estimated K from its actual value resulted in the relative error ranges from 4 % to 12 % in the computed DEL. Another way is to compare the inherent frequencies of the estimated load with the actual one as done in Fig. 8. As seen in Fig. 8, all the dominant mode frequencies are captured in the estimated load.

25) Page 11, line 228 –"At this point, it is important to remember that the estimated K value is a combination of the main shaft and blade edgewise stiffness." What does this imply for the reader? How does this change our interpretation of findings? If other components are present, how large are they relative to shaft K. Can you perform a simple analysis be taking blade edge stiffness values from the HAWC2 model and comparing with a known value of shaft torque to demonstrate that the dominant contribution to K is from the shaft?

Response:

The drivetrain torsional frequencies of the DTU 10 MW for two different modeling assumptions (without blade edge stiffness and with it) are presented [11] from here.

|  | When blade flexibilities included | Rigid blade (i.e., only the main shaft is flexible) |
|---|---|---|
| Drivetrain Torsional frequency (Hz) | 1.8 | 4.003 |

As seen in the above table, the addition of blade flexibilities resulted in a 50% reduction in the drivetrain torsional frequency. This is due to the fact that the rotor inertia is significantly higher than the drivetrain components, [For DTU 10 MW turbine [11], rotor inertia, Jr= 8.6e07 kgm^2, generator inertia, Jg=3.75e06 kgm^2; all the inertia values about the low-speed side].This is not the case for the gearbox dynamics, since the inertia of the gearbox is approximately a factor of 1 /30 of the generator inertia [4]. Hence, these components do not significantly affect the main shaft torsional frequency.

26) Page 11, line 231– Again,static and dynamic theta values need to be introduced much earlier in the paper, and described without assuming familiarity. Also the method by which static theta is estimated should form part of a methodology section, rather than being introduced in amongst results.

Response:

As per reviewer's suggestion, the definition and the estimation of the static component will be given in Section 2.

*27)* Page 11, line 233 –What about gearbox efficiency? If generator efficiency is taken into account won't gearbox efficiency also have an effect? If not then please outline why, e.g. maybe the efficiency of one if much higher than the other? Otherwise both should be accounted for.

Response:

In the article, the generator efficiency is taken for the static calculations. However, as an alternative approach, one can use the overall efficiency of the drivetrain as well which accounts for the efficiencies of all components for completeness. This does not significantly change the results. The use of the generator efficiency resulted in the offset of 2.5 % about the mean value of the rotor torque in the estimated static component of theta. This explanation will be added to the revised manuscript.

*28)* Page 12 – Damage equivalent load calculations: The Wöhler exponent takes on different values, which is used here? Does hollowness of a shaft have an effect to these calculations?

Response:

The Wöhler exponent for the cast iron of m=6 [10] is used in the study and the effect of shaft hollowness is not considered.

*29)* Page 13, line 254 – "This could be due to fact that the pitch control will be active beyond the rated wind speed and hence the instantaneous changes in the pitch angle affects the rotor speed." Isn't this a bit of a gap in the modelling? Wind turbines spend a fair amount of time operating while pitching so this is an important factor. This all ties to the approximations being made when the chosen model is applied. A better discussion of the model in relation to a real world turbine is needed when the model is presented, and this must include effects such as pitch that are accounted for, along with some indication of the possible impact that will have. E.g. "The proposed model does not account for shaft speed changes related to pitch activity, however, in above rated conditions the controller is acting to maintain a constant rotational speed, and so such changes (and associated errors) are expected to be small…" or something like that.

Response:

The fact that the peak amplitudes are not captured in the reconstructed torsional displacements is typical for Tikhonov regularisation as there is a slight mismatch in the frequency spectra between the actual and reconstructed torsional oscillations [15]. Further, the pitch angle influences the main shaft torsional oscillation through the rotor torque and the rotor speed. However, the controller is maintaining a constant rotational speed beyond the rated wind speed, hence the instantaneous changes in the pitch

angle are not propagated through the rotor speed to the main shaft oscillations. All these effects, in addition to the uncertainty in K estimation, resulted in a maximum error of 12 % in the estimated DEL.

This explanation will be added to the revised manuscript.

*30)* Page 13 (and onwards) – You start referring to "1Hz DELs" without defining what these are.

Response:

When Nref in Eq. (21) equals the component's lifetime in seconds, then the DEL has the frequency of 1 Hz. Since Nref of 600 cycles is used in the preset study, it becomes 1 Hz DEL over a 10 min period. This definition will be added to the revised manuscript.

*31)* Page 13, line 258 – "SCADA measurements of the Vestas V52-850 kW research turbine..." What frequency of data?

Response:

Throughout this article, the input data has a sampling frequency of 50 Hz.

The concerned sentence will be modified as follows:

In particular, measurements taken during January 2019 consisting of 4459 ten minute recorded cases with 50 Hz sampling frequency were used for this study.

*32)* Page 14, lines 278-282 – You make big claims about remaining useful life, but only actually calculate damage equivalent loads. Unless you demonstrate how this link is made (which is non-trivial) you maybe need to make the claims less strong. E.g. `the proposed methodology may assist in decision making regarding maintenance and remaining useful life' kind of thing. Or, reference to material that shows how damage equivalent load predictions are then made into remaining useful life assessments. But either way, claims need to be tentative unless backed up by references or evidence.

Response:

The estimated main shaft torsional DEL may be used to quantify the  RUL of the drivetrain [12]. However, this is beyond the scope of the current work. The concerned sentence will be modified accordingly.

*33)* Page 15, line 281 – "It is also feasible to reconstruct the torsional moment time series, which may be used an input to gearbox design tools to predict the loading within the gearbox." This is another big claim, especially considering that you have assumed a perfectly stiff gearbox with 100% efficiency, if included it needs to be demonstrated more fully that this is indeed the case. It is also important to consider what you have shown, which is that damage equivalent loads are well predicted by your method, that is not necessarily the same as saying that the time-series loads are reconstructed to a similar level of accuracy. They may be, but that needs discussion and consideration and doesn't follow immediately from the damage estimate results.

Response:

As per the new modelling assumption, the estimated stiffness is the collective stiffness of the entire drivetrain since the inputs are from the measurements. Hence, the estimated torsional load can be used as an input to gearbox design tools.

Regarding the reconstructed torsional load time series, all the important aspects of the time-series variation and the dominant frequency dynamics (low-frequency components - up to first three peaks ) are captured quite well in the reconstructed torsional load as compared to the simulated one as shown in Fig. 8 of the manuscript. Hence, it is established in the article that both the torsional load and the torsional DEL can be reconstructed with  reasonable accuracy. However, to be consistent with the terminology, the torsional moment time series will be changed as torsional load time series.

*34)* Page 15, lines 297-298 – "As the proposed methodology does not require any design basis, it can be used for any older turbines for the estimation of the main shaft torsional RUL." This is not necessarily true. As mentioned, older turbines will at best tend to have 10min averaged SCADA data available. What does this mean for your methodology if that's the case? If high-frequency SCADA is always required then that needs to be made much clearer throughout the paper, and probably indicates that the term `high-frequency SCADA' should be used in the paper title, since most people assumed SCADA = 10min averages. The methodology is still potentially relevant to older turbines since new data can be collected to link fatigue damage to its operating conditions, but life estimates then require some sort of hindcasting or statistical analysis that goes beyond what is covered here. Again, please consider which claims are reasonable to make based on what is demonstrated in the paper, if reaching beyond what is presented here then please add more discussion and references which evidence the claims being made.

Response:

The concerned sentence has been modified based on the findings of the current work. The modified sentence reads:   As the proposed methodology does not require any design basis or an aeroelastic

design basis, it can be used for wind turbines that possess high-frequency SCADA measurements for the estimation of the main shaft torsional load and DEL.

Please refer to the previous discussions (i.e., response to query 5) about the sampling frequency of the SCADA measurement. In addition, the phrase 'high-frequency SCADA' will be included in the title as per the reviewer's suggestion.

**References:**

[1] Boukhezzar, B., Lupu, L., Siguerdidjane, H., & Hand, M. (2007). Multivariable control strategy for variable speed, variable pitch wind turbines. Renewable energy, 32(8), 1273-1287.

[2] Shin, Y. H., Moon, S. J., Kwon, J. I., & Chung, T. Y. (2013). Derivation of 4 degrees of freedom nonlinear wind turbine model using effective mass and stiffness for simulation of control algorithm. Journal of Renewable and Sustainable Energy, 5(5), 052012.

[3] Berglind, J. B., Wisniewski, R., & Soltani, M. (2015, July). Fatigue load modeling and control for wind turbines based on hysteresis operators. In 2015 American Control Conference (ACC) (pp. 3721-3727). IEEE.

[4] Novak, P., Ekelund, T., Jovik, I., & Schmidtbauer, B. (1995). Modeling and control of variable-speed wind-turbine drive-system dynamics. IEEE Control Systems Magazine, 15(4), 28-38.

[5] Singh, M., & Santoso, S. (2011). Dynamic models for wind turbines and wind power plants (No. NREL/SR-5500-52780). National Renewable Energy Lab. (NREL), Golden, CO (United States).

[6] Girsang, I. P., Dhupia, J. S., Muljadi, E., Singh, M., & Pao, L. Y. (2014). Gearbox and drivetrain models to study dynamic effects of modern wind turbines. IEEE Transactions on Industry Applications, 50(6), 3777-3786.

[7] Pahn, T. (2013). Inverse load calculation for offshore wind turbines (Doctoral dissertation, Hannover: Gottfried Wilhelm Leibniz Universität Hannover).

[8] Pahn, T., Rolfes, R., & Jonkman, J. (2017). Inverse load calculation procedure for offshore wind turbines and application to a 5-MW wind turbine support structure. Wind Energy, 20(7), 1171-1186.

[9] Lapidus, L. and Pinder, G. F., Numerical Solution of Partial Differential Equations in Science and Engineering], John Wiley & Sons, New York (1982).

[10] Veldkamp, H. F. (2006). Chances in wind energy: A probalistic approach to wind turbine fatigue design (Doctoral dissertation, Faculty of Mechanical Maritime and Materials Engineering, TU Delft).

[11] Bak, C., Zahle, F., Bitsche, R., Kim, T., Yde, A., Henriksen, L.C., Hansen, M.H., Blasques, J.P.A.A., Gaunaa, M. & Natarajan, A., 2013. The DTU 10-MW reference wind turbine. In Danish Wind Power Research 2013.

[12] Pagitsch, M., Jacobs, G., & Bosse, D. (2020). Remaining Useful Life Determination for Wind Turbines. In Journal of Physics: Conference Series (Vol. 1452, No. 1, p. 012052). IOP Publishing.

[13] A Brandta and R. Brincker, Integrating time signals in frequency domain – Comparison with time domain integration, Measurement 58 (2014) 511-519.

[14] Qihe, L. (2019, November). Integration of vibration acceleration signal based on labview. In Journal of physics: conference series (Vol. 1345, No. 4, p. 042067). IOP Publishing.

[15] Lee, H. S., Hong, Y. H., & Park, H. W. (2010). Design of an FIR filter for the displacement reconstruction using measured acceleration in low-frequency dominant structures. International Journal for Numerical Methods in Engineering, 82(4), 403-434.

---

## Author Comment (AC2)

**Identification of wind turbine main shaft torsional loads from high-frequency SCADA measurements using an inverse problem approach**

The authors thank the reviewer for the valuable comments that improve the article quality. We have addressed all the reviewer's queries in detail in the following:

**Reviewer 2**

**General comments**

1) The first question concerns the simplified mathematical model of the drivetrain (Eqs 1-3). This model is a compact two-disk representation, in which the inertia of the main shaft and that of the gear box and high-speed shaft are collected into the equivalent generator inertia. Therefore, for the sake of consistency of the model, the stiffness term (K) seems to stand for the collective stiffness of the whole drivetrain rather than solely that of the main shaft [1]. However, the manuscript is taking whole drivetrain apart from the main shaft perfectly rigid, so above stiffness is assigned to the main shaft and will then be identified and utilised it to infer the main shaft loads, which may not be right for drivetrains with multi-stage gearbox and shafts. There are several references available (e.g. [2]) that explain the equivalent drivetrain models as the combination of springs in series. The angular displacement of such models are basically the summation of the individual displacements of the drivetrain components. A comparison with simulation results cannot simply validate the above assumption, as the turbine simulation tools are also usually including a simplified drivetrain model, as they aim to capture the global motion and loads of system. In case that the contribution of the other components are known/found to be negligible, this has to be quantitatively demonstrated or at least be discussed adequately. It is not well described whether this work is identifying the main shaft load or the total equivalent torsional load of the drivetrain. Nevertheless, the manuscript's methodology and findings are still interesting. As such, it is recommended to either clarify this point or discuss the probable limitations.

Response:

The two-mass model is widely used to study the torsional oscillations of the mechanical drivetrain of a wind turbine [1-5] because it captures the underlying free-free torsional frequency. This is the main driving frequency for the main shaft torsion. Though the governing equations (i.e., Eqs. (1-3) of the manuscript) remain the same for all the considered work. The underlying assumption regarding the stiffness of the various components in the drivetrain is different for each model and the definition for the 'K' in Eqs. (1-3) in each of these works is given in the following table:

| Model | Considered stiffnesses for K |
|---|---|
| Boukhezzar et al. [1] | Rotor, generator and the main shaft |
| Shin et al. [2] | Main shaft |
| Berglind et al. [3] | Collective stiffness with rigid gearbox |
| Novak et al., [4] | Main shaft |
| Singh & Santoso, [5] | Main shaft and high-speed shaft, gearbox dynamics dynamic neglected |
| Girsang et at., [6] | Stiffness of all components were considered. |

As seen in the table, the modelling assumptions regarding the gearbox and main shaft are inspired from [1-5]. However, for the inverse-problem based approach, the inputs to the drivetrain model (i.e., the rotor and generator speed) determine the contribution of each component to the 'K' value. For example, if the inputs are from the HAWC2 aeroelastic simulation then the estimated stiffness value has the contributions from the rotor and the main shaft only since the simulation does not account for the gearbox dynamics. If the inputs from the measurements are used, then the estimated stiffness is a collective stiffness that has a contribution from all the components.

To be consistent with all the inputs, the modelling assumptions adopted by Girsang et al.[6] has been referred in the revised manuscript (i.e., by considering the contributions from all the components in the drivetrain). Nevertheless, these modelling assumptions will not affect the methodology proposed herein.

The concerned sentences (i.e., page 3, line 86 and line 88 and Page 4, line 98 and 99) will be removed from the revised manuscript. Accordingly, the discussion about the inputs to the model will be included along with the reference (Girsang et al., [6]) in the revised manuscript. Since the modelling assumption is revised then the concerned figure (Fig. 2 of the old manuscript) does not add any value to the paper and hence it will be removed from the manuscript.

2) The other point is regarding the manuscript's argument that it just needs SCADA data, which sounds compelling due to this data being readily available. The suitable sampling rate of the SCADA data that is required for this methodology has not been discussed within the paper and needs to be closely clarified. This point becomes more important when the manuscript argues that their findings are beneficial to the calculations for life extension of wind turbine system. However, many of the existing turbines that would require life extension are equipped with SCADA systems with very low frequency output data, normally averaged values of order of minutes. It seems that this drawback is going to be removed by artificially increasing the SCADA signal sampling rate without providing any information on the original and the resulted sampling rate. Therefore, it is a far question whether this methodology is really applicable to SCADA data

or condition monitoring system's data is still required. As such, it is again recommended to either clarify this point or discuss the probable limitations.

Response:

We thank the reviewer for pointing out this fact. The proposed approach requires that the sampling frequency of the SCADA measurement be significantly higher than the dominant frequencies of the drivetrain torsional oscillations (i.e., 1p and 3p rotor excitation frequencies and torsional natural frequencies). As a result, the proposed method cannot be used for the turbines that have measurements in terms of 10-min SCADA statistics. The larger the turbine, the lower are the dominant frequencies of the torsional oscillations, hence for large size turbines the sampling frequency of SCADA measurements can be lower. In addition, we have used the phrase high-frequency SCADA in the article title as well. Also, the authors are neither proposing any method for the new data collection nor resampling the measurement data. It is demonstrated that with the existing high-frequency SCADA measurements, the site-specific torsional load can be estimated. This can be used to estimate the yearly damage without historical weather records or condition monitoring data as the wind speed and wind direction measurements are available in the SCADA measurements.

Also, the arguments regarding the estimation of the RUL have also changed to emphasize that those are not the scope of the current work.

We have rewritten the concerned paragraph of the introduction as follows:

The estimated shaft torsional stiffness and displacement are further used to identify the site-specific shaft torsional load. This novel methodology can potentially benefit wind farm owners since both the property of the structure in terms of its stiffness and the structural response and the site-specific load can be determined without requiring additional sensors or information from the wind turbine manufacturer. The significance of estimating the main shaft torsional load is that it affects the fatigue performance of other drivetrain components such as gearbox and planetary bearings (Dong et al., 2012; Gallego-Calderon and Natarajan, 2015; Ding et al., 2018). Hence, the same site-specific torsional load may be used as an input for quantifying the remaining useful life (RUL) (Ziegler et al., 2018) of the main shaft, gearbox, and other drive train components as well. The estimation of RUL/ yearly damage does not require additional historical weather data and condition monitoring data as the wind speed and wind direction measurements are available in the SCADA measurements. However, this is beyond the scope of present work. Further, the proposed approach requires that the sampling frequency of the SCADA measurement be significantly higher than the dominant frequencies of the drivetrain torsional oscillations (i.e., 1p and 3p rotor excitation frequencies and torsional natural frequencies). As a result, the proposed method cannot be used for the turbines that have measurements in terms of 10-min SCADA statistics.

3) Moreover, there are well-established integration techniques that are being used, even for online conversion of acceleration to velocity and displacement signals [3, 4]. In the absence of artificial noise in the simulated speed data, the type of drift shown in Figures 3 and 4 are apparently showing a linear trend due to the accumulation of the error from the initial value. This type of trend can be usually avoidable by the common digital filtering with restriction of frequency range within the pre-processing of the signal, particularly when the signal's mean value (static term) is going to be added later on separately, in which case the initial value of integration doesn't really matter. As such, the authors need to mention that the application of regularisation within a trapezoidal scheme is not the only available method to stably convert velocity into displacement.

Response:

Digital filters and frequency domain integration approach (FDIA) are the widely used techniques in literature to reconstruct displacements from the measured accelerations [7-9]. However, digital filters such as impulse response filters (IIR) and finite response filters (FIR) have several drawbacks when reconstructing the low-frequency displacements, as is the case here [9]. On the other hand, the FIDA methods are sensitive to the time intervals of the measurements [9]. Hence, a least square minimization based regularization technique called Tikhonov regularisation is used in the present study as it is better suited for low-frequency dominant structures as shown by [9].

The above discussion will be added into the introduction of the revised manuscript.

**Minor comments:**

1) The derivatives for the inverse integration scheme is similarly given in Hong et al 2008, so please refer to this paper in the beginning of the corresponding section.

Response:

The reviewer's suggestion will be incorporated into the revised manuscript.

2) Paragraph 15: "and" missing after strain gauges.

Response:

The reviewer's suggestion will be incorporated into the revised manuscript.

3) Paragraph 75: This is questionable if older turbines really possess SCADA with a desirable sampling rate that suits the manuscript's methodology.

Response:

Please refer to the response to the general comment 2.

4) Paragraph 85: As discussed previously, the assumption of the fully rigid gearbox and higher stage shafts, particularly for a multi-stage systems, needs further clarifications.

Response:

Please refer to the response to the general comment 1.

5) Paragraph 190: computationally "more" expensive.

Response:

The reviewer's suggestion will be incorporated into the revised manuscript.

6) Paragraph 205: Please clarify those three yaw directions.

Response:

The reviewer's suggestion will be incorporated into the revised manuscript.

7) Paragraph 210: where is the noise coming from that needs to be damped. Please clarify.

Response:

There are two sources of the noise: (i) numerical noise and (ii) measurement noise. This clarification will be added in the concerned place.

8) Paragraph 220: "the" modes.

Response:

The reviewer's suggestion will be incorporated into the revised manuscript.

9) Paragraph 225: pds should be expanded.

Response:

The explanation for pdf (i.e, probability density function)  will be given in the revised manuscript.

   10) Mixed use of symbols has to be avoided throughout the manuscript.

Response:

The reviewer's suggestion will be incorporated into the revised manuscript.

   11) Eq 18, it seems that gearbox ratio is missing somewhere.

Response:

The gearbox ratio is used when converting the generator torque from the high-speed side to the low-speed side. $T\_g$ in Eq. (19) refers to the generator torque on the low-speed side which already accounts for the gearbox ratio. This explanation of the generator torque will be given in the problem formulation section of the revised manuscript.

   12) Fig 8, it seems that some constant or near constant offset exist (observable in both time and frequency plots), apparently theta_static is not appropriately identified. The plotted FFT is too noisy, one can use a better illustration, perhaps, a power spectrum with sufficient averaging to get more clear peaks, as at the moment it is barely possible to get a good picture of the main peaks.

Response:

The constant offset is due to the use of the generator efficiency for all the static displacement calculations. As a result, there is an offset of 2.5 % about the mean value of the rotor torque. However, as an alternative approach, one can use the overall efficiency of the drivetrain as well which accounts for the efficiencies of all components for completeness.

 Power spectral density (PSD) plots will be given in the revised manuscript instead of the FFT plots.

   13) Paragraph 20: what are the sampling rate details and how the SCADA data's frequency resolution was increased, is it practically legitimate? Please clarify.

Response:

The sampling frequency of the SCADA measurement is 50 Hz. In the considered work, the authors have not increased the sampling rate of the measurement.

14) **The results using the actual data does not look to add substantial value to the manuscript, as it is not really supporting the validation of the methodology.**

Response:

The authors feel that including the results from the V52 measurements shows that the resulting trend is very similar to the trend observed from the HAWC2 results. Since RUL prediction on wind farms is based on the margins available on the actual structure as compared to prior aeroelastic computations, this result demonstrates the confidence in the proposed method to be utilized in RUL computations of wind farms.

**References:**

[1] Boukhezzar, B., Lupu, L., Siguerdidjane, H., & Hand, M. (2007). Multivariable control strategy for variable speed, variable pitch wind turbines. Renewable energy, 32(8), 1273-1287.

[2] Shin, Y. H., Moon, S. J., Kwon, J. I., & Chung, T. Y. (2013). Derivation of 4 degrees of freedom nonlinear wind turbine model using effective mass and stiffness for simulation of control algorithm. Journal of Renewable and Sustainable Energy, 5(5), 052012.

[3] Berglind, J. B., Wisniewski, R., & Soltani, M. (2015, July). Fatigue load modeling and control for wind turbines based on hysteresis operators. In 2015 American Control Conference (ACC) (pp. 3721-3727). IEEE.

[4] Novak, P., Ekelund, T., Jovik, I., & Schmidtbauer, B. (1995). Modeling and control of variable-speed wind-turbine drive-system dynamics. IEEE Control Systems Magazine, 15(4), 28-38.

[5] Singh, M., & Santoso, S. (2011). Dynamic models for wind turbines and wind power plants (No. NREL/SR-5500-52780). National Renewable Energy Lab. (NREL), Golden, CO (United States).

[6] Girsang, I. P., Dhupia, J. S., Muljadi, E., Singh, M., & Pao, L. Y. (2014). Gearbox and drivetrain models to study dynamic effects of modern wind turbines. IEEE Transactions on Industry Applications, 50(6), 3777-3786.

[7] A Brandta and R. Brincker, Integrating time signals in frequency domain – Comparison with time domain integration, Measurement 58 (2014) 511-519.

[8] Qihe, L. (2019, November). Integration of vibration acceleration signal based on labview. In Journal of physics: conference series (Vol. 1345, No. 4, p. 042067). IOP Publishing.

[9] Lee, H. S., Hong, Y. H., & Park, H. W. (2010). Design of an FIR filter for the displacement reconstruction using measured acceleration in low-frequency dominant structures. International Journal for Numerical Methods in Engineering, 82(4), 403-434.

---

## Author Response (AR1)

**Identification of wind turbine main shaft torsional loads from high-frequency SCADA measurements using an inverse problem approach**

The authors thank the reviewers for their valuable comments that improve the article quality. We have addressed all the reviewers' queries in detail in the following:

**Reviewer 1:**

**General comments**

**Structure, clarity and readability** — in its current form the paper is haphazard and difficult to follow. The introduction jumps between background information and descriptions of the methodology in the current work. The structure and flow of the paper overall is not clear, with the reader often left wondering where a given section is headed, or having to try and work out whether a certain concept is being newly introduced or if the author is referring to something already discussed. The paper needs to be reorganised such that introduction, background, methodology and results are all clear and distinct from one another — with the overall structure and flow of the paper made clearer from the start, and better signposted on the way through.

**Modelling and analysis presentation and associated discussions** — Descriptions of the modelling and analysis undertaken in this paper is currently lacking in clarity, completeness and context. The lumped parameter model is introduced without discussion of the approximations being made and the potential impacts this could have on model accuracy. Inputs to the model are not described in sufficient detail, e.g., frequency requirements for input SCADA data applied throughout this work, and the effect data frequency may have on the presented method, are not properly discussed. Note, when SCADA data is mentioned, one normally assumes 10 min mean values are being referred to, whereas high frequency SCADA (50Hz) is used here for at least part of the analysis. This is therefore a possible source of confusion for readers of the current paper. Also, the stiffness value being approximated is stated to contain contributions from components other than the main shaft, again the effect of this (good/bad/neutral) is not discussed, the fact is merely stated. The descriptions of the Collage method and Tikhonov regularisation, in their current form, are not appropriate. Much of the mathematics is simply stated without proper definitions for the various terms, or suitable descriptions to ensure the reader can follow what is being said. Notation also needs to be revised, with the same symbols currently representing both scalar and vector quantities. While I applaud a proper treatment of the mathematics, this is only useful if the formulations can be followed and reproduced by the reader. This requires careful definitions for all terms and better descriptions of how the resulting algorithms are implemented. I suggest that the authors consider providing a shorter description of each methodology, including why it is necessary and roughly how it works (e.g. provide an analogy), along with the resulting equation to be solved (e.g. Equation 6 for the collage method) and the solution method used. Then, the full

mathematical details can be referred to an appendix where they can be fully and carefully described, without derailing the flow of the overall paper.

**Context, motivations and claims** – While undoubtedly useful to the wind energy field, context regarding the wind industry is somewhat lacking from the paper in its current form. For example: is main-shaft fatigue currently of concern, are there available failure rates for this component, has it been much researched in the literature to date? As mentioned above this work seems to assume high frequency SCADA data for its analysis, but that isn't necessarily standard, especially for older turbines. The paper claims that the method is useful for older turbines, but what if they only have 10-min SCADA available? The paper shows certain levels of error for damage equivalent loads estimates, what does that accuracy level mean in terms of usefulness in the field or an ability to predict remaining useful life? Claims are made regarding remain useful life, but the link to that from results in the paper is non-trivial given variability of conditions and potentially incomplete histories of operating data – statistical analysis would likely be required. Note the above points don't necessarily all have to be bottomed out fully, but they should be discussed at some level in this work.

Response:

The authors have restructured the article by addressing the reviewers general and specific comments. By addressing the specific comments, the authors feel that the general comments will also be addressed. To address the reviewer's general and specific comments that concerns the introduction, the same has been re-written in the revised manuscript. Further, the detailed mathematical formulation of  the Tikhonov regularisation and the Collage method has been moved to Appendix and the resulting equation along with its usage alone is discussed in the main part of the paper.

**Specific comments**

1) *Page 2, lines 30-44* - why list all of these methods if you're not going to discuss them? Beyond stating that they exist its necessary to tell the reader something about them, e.g. pros, cons, what's relevant to the current study, why you used the method you did instead of these.

Response:

The concerned sentences that list various OMA methods have now been removed. Instead, studies that use such OMA methods for wind turbine system identification are listed with appropriate references that discuss their pros and cons.  Hence, we have not repeated the pros and cons of these methods herein. These existing methods require the system response *a priori* for identifying the modal parameters. Usually, the system response is measured using accelerometers and strain gauges. The addition of new instrumentation to existing turbines, such as the installation of strain gauges, accelerometers can be

costly and require repetitive calibration and synchronization of their measurement signals with the turbine computer. Hence, an alternative cost-effective approach is proposed here to estimate both the main shaft structural response and modal parameters directly from high-frequency SCADA measurements without using any additional measurement sensors. We have rewritten the concerned parts of the introduction. Please refer to para 2 and 3 of the introduction in the revised manuscript.

2) Page 2, lines 42-43 – "However, to the best of the authors' knowledge, there is no such study available on estimating the structural response of a wind turbine component from SCADA measurements." You mentioned previously that many studies exist which consider OMA for wind turbines, do these not use SCADA data at all? Maybe describe them in more detail or list what they do use to make this claim easier to interpret.

Response:

The concerned sentences (i.e., L 42-43 of the reviewed manuscript) have been removed in the revised article. However, as mentioned in the previous reply, the existing OMA methods use measured structural response for the identification of the modal parameters, whereas the proposed method uses high-frequency SCADA measurement to estimate both the structural response (torsional displacement) and the modal parameters (Shaft stiffness).

3) Page 2, lines 45-47 - We're now into methodology and haven't even reached the problem formulation yet. This is why splitting into intro, background, methodology seems to make sense. You can then give the summary overview of the types of models to be used in the intro, and then just go straight to the literature in background, that makes the flow feel better.

Page 3, line 60 - Again, this should be in Background - well in the summary of the methods at least, wherever that ends up being.

Response:

Estimating the main shaft torsional dynamic response and properties from high-frequency SCADA measurements is the scope of the work. This is done using an inverse problem-based approach that consists of Tikhonov regularisation for regularising the measurement data and the collage method for the system identification. As per the reviewer's suggestion, the introduction part concerning the methodology has been rewritten in such a way that the problem formulation is followed by an explanation of the methodology. Please refer to the introduction of the revised manuscript.

4) Page 3, lines 73-74 – "Hence the same site-specific torsional load can also be used to quantify the RUL of gearbox and other drive train components as well". These are very big claims that need to be demonstrated if they are being made in the paper. Also in your drivetrain model don't you

assume the gearbox is completely stiff? Does that not impact your ability to make conclusions regarding the gearbox? These are some of the reasons why I feel a better discussion of the applied model and its underlying assumptions is necessary.

Response:

Studies by Dong et al., 2012; Gallego-Calderon and Natarajan, 2015; Ding et al., 2018 showed that the torsional load of the main shaft can significantly affect the fatigue performance of the drivetrain components such as gearbox bearings. Hence, the estimated site-specific main shaft torsional loads using our proposed methodology may be used as an input for quantifying the RUL of the other drivetrain components. However, the specific quantification of the RUL of the gearbox is beyond the scope of the current work. The assumption regarding the drivetrain model will be revised as per the reviewers' suggestions. Accordingly, if the input data (i.e., high-frequency SCADA) accounts for the gearbox flexibility, then the estimated torsional load also has the contribution from the gearbox. This will be discussed in more detail while addressing the query (query 6) about the modelling assumption.

We have rewritten the concerned paragraph in the manuscript. Please refer to the introduction (para 7) of the revised manuscript.

5) Page 3, lines 74-76 – Older turbines will likely not have a full service history of 1Hz SCADA data, how is the method applied in that case? Are you proposing that new data can be collected and used to estimate yearly damage for those turbines? That would also need some linking to historical weather records I'd imagine. This is not a trivial claim to make so more discussion is necessary.

Response:

We thank the reviewer for pointing out this fact. The proposed approach requires that the sampling frequency of the SCADA measurement be significantly higher than the dominant frequencies of the drivetrain torsional oscillations (i.e., 1p and 3p rotor excitation frequencies and torsional natural frequencies). As a result, the proposed method cannot be used for the turbines that have measurements in terms of 10-min SCADA statistics. In addition, we have used the phrase high-frequency SCADA in the article title as well. Also, the authors are not proposing any method for the new data collection. With the existing high-frequency SCADA measurements, the site-specific torsional load can be estimated. This can be used to estimate the yearly damage without historical weather records as the wind speed and wind direction measurements are available in the SCADA measurements.

We have rewritten the concerned paragraph in the manuscript to incorporate the above explanation. Please refer to the re-written introduction (para 7).

6) Page 3, line 86 – "gearbox is perfectly stiff". This approximation, among others, needs discussion and perhaps references, is it reasonable to assume this? What are the likely errors from it? Do any other papers deal with this type of approximation and show its effect?

Page 3, line 88 – "The main shaft is modelled by an inertia free viscously damped torsional spring". Same as for gearbox, the validity and effect of these approximations is necessary to allow the reader to understand how this model relates to the real world.

Page 4, line 98 – "the high frequency gearbox dynamics do not play a significant role in it." Do you have a reference for this claim? It needs to be demonstrated, referenced or at least discussed.

Page 4, line 99 – "Hence, the two-mass model is sufficient enough to model the shaft torsional dynamics for the wind turbine normal operations as it includes both the low frequency modes" This is a claim, can you back it up with evidence?

Response:

The two-mass model is widely used to study the torsional oscillations of the mechanical drivetrain of a wind turbine [1-5] because it captures the underlying free-free torsional frequency. This is the main driving frequency for the main shaft torsion. Though the functional form of the governing equations (i.e., Eqs. (1-3) of the manuscript) remain the same for all the considered work, the underlying assumption regarding the stiffness of the various components in the drivetrain is different for each model and the definition for the 'K' in Eqs. (1-3) in each of these works is given in the following table:

| Model | Considered stiffnesses for K |
| --- | --- |
| Boukhezzar et al. [1] | Rotor, generator and the main shaft |
| Shin et al. [2] | Main shaft |
| Berglind et al. [3] | Collective stiffness with rigid gearbox |
| Novak et al., [4] | Main shaft |
| Singh & Santoso, [5] | Main shaft and high-speed shaft, gearbox dynamics neglected |
| Girsang et at., [6] | Stiffness of all components were considered. |

As seen in the table, the modelling assumptions regarding the gearbox and main shaft are inspired from [1-3]. However, for the inverse-problem based approach, the inputs to the drivetrain model (i.e., the rotor and generator speed) determine the contribution of each component to the 'K' value. For example,

if the inputs are from the HAWC2 aeroelastic simulation then the estimated stiffness value has the contributions from the rotor and the main shaft only since the simulation does not account for the gearbox dynamics. If the inputs from the measurements are used, then the estimated stiffness is a collective stiffness that has a contribution from all the components including gearbox.

To be consistent with all the inputs, the modelling assumptions adopted by Girsang et al.[6] has been referred in the revised manuscript (i.e., by considering the contributions from all the components in the drivetrain). Nevertheless, these modelling assumptions will not affect the methodology proposed herein.

Further, the main reason for ignoring the gearbox dynamics in some of the studies is that the torsional oscillation of the main shaft is mainly dominated by low-frequency components and hence the high-frequency gearbox dynamics will not significantly alter the main shaft torsional oscillations [4,5].

The concerned sentences (i.e., page 3, line 86 and line 88 and Page 4, line 98 and 99 of the reviewed manuscript) is removed in the revised manuscript. Accordingly, the discussion about the inputs to the model is included along with the reference (Girsang et al., [6]) in the revised manuscript. Please refer to L 96-99 and L 103-105, P 4.

7) Page 4, line 100 – "Also, given the system parameters and rotor and generator torques, the two-mass model is capable of predicting the shaft torsional displacement as close as that of the full-fledged aeroelastic simulation as shown in Fig. 2." Another un-evidenced claim. Please discuss.

Response:

Since the modelling assumption has now been revised in the manuscript, the concerned figure does not add any value to the paper and hence it is removed from the revised manuscript.

8) Page 4, line 102 – HAWC2 needs better description, what does this include (BEM model + multibody dynamics? etc.). Not all readers will be familiar with this code.

Response:

HAWC2 (Horizontal Axis Wind turbine simulation Code 2nd generation) is an aeroelastic simulation tool that is used for calculating the wind turbine loads and responses in time domain. It uses multibody formulation to model the structure, blade element momentum (BEM) theory-based models for modeling the wind effects and hydrodynamic models for modeling the hydro-effects (in the case of offshore turbines) on the structure. Control of the turbine is performed through the dynamic link libraries (DLLs).

The above description is added to the revised manuscript in addition to the existing reference for the HAWC2. Please refer to L 151-155, P 7.

9) Page 4, line 104 – "In forward problem approach" You are using lots of terminology that has not been defined, don't assume everyone knows what this means.

Response:

The concerned sentence is modified with a reference as follows:

Given the modal parameters $(J_r, J_g, C, K)$ and external torques $(T_r$ and $T_g)$, Eqs (1-3) are solved for $\omega_r$, $\omega_g$ and $\theta$, which is known as a forward problem approach [7]. Please refer to L 107-108, P 4.

10) Page 4, line 109 - How do you differentiate the theta values? Finite difference can go wrong for 'noisy measurements', do you apply a filter?

Response:

It is indeed true that the finite difference methods are highly sensitive to measurement noise. For this purpose, a regularisation technique called the Tikhonov regularisation was used in the study. It minimizes the error due to numerical and measurement noise using the least-square criteria by means of numerical damping. The implementation of this regularisation technique is discussed in the latter part of the paper. Refer to Section 2.1 of the revised manuscript.

11) Page 5– see comments in 'General' concerning presentation of mathematical methods in the paper. Too high a burden us placed on the reader here, you need to decide what to keep, what to add and what to move to an Appendix. Currently readers would struggle to recreate these methods from the descriptions provided and the overall paper gets bogged down in technical details which are important but probably best in an Appendix and with improved definitions and descriptions.

Response:

Only a short description of the collage method and Tikhonov regularization along with the resulting equations is presented in the methodology section. The detailed mathematical presentation about the collage method and the Tikhonov regularization is moved to the Appendix as per the reviewer's suggestion. Please refer to Section 2.1, 2.2 and Appendix A.1 and A.2 of the revised manuscript.

12) Page 6, Equation 6– is there an issue here? Should the 'u' in the integral operator now be 'theta'?

Response:

The authors thank the reviewer for pointing out the typo error. The authors have now given the correct equation in the revised manuscript. Please refer to Eq. (5) in P 6.

13) Page 6, line 137 – "hence it is important to check" This sentence does not necessarily hold. Just because something hasn't been tried it doesn't mean it should. There should be a better justification than that alone.

Response:

The concerned sentence is modified as follows:

To test the applicability and efficiency of the collage method for the wind turbine drivetrain system, a verification study is undertaken by comparing the modal parameters obtained using the collage method with its design value for two different wind turbines.  Please refer to L 145-147, P 7.

14) Page 6, line 150– "As explained earlier," by this point the earlier descriptions have all merged with other details, another reason to have a clear, concise description of the methodology somewhere which is easily remembered and referred back to. Remember readers might not be familiar with these types of method so much will be new to them, some of these descriptions feel like they assume quite a high level of familiarity. Also I'm not sure 'time integration' was specifically mentioned earlier in the paper.

Response:

The concerned sentence is modified as follows:

Given $\omega_r$ and $\omega_g$, $\dot{\theta}$ is obtained by using Eq. (3) and by numerically integrating the $\dot{\theta}$, the shaft torsional displacement ($\theta$) is obtained. Please refer to L 115-116, P 5. Also, as per the reviewer's suggestion, only the concise description of the methodology is presented in Section 2.

15) Page 7, Figure 3 – It is not clear to the reader why the numerically integrated estimate of theta diverges here. A 'lack of initial conditions plus measurements noise' is given as a reasons, but that doesn't help explain why. Is it because noise leads to greater 'areas under the curve' when integrating? But I'd expect that effect to give both higher and lower values. Please give a more detailed description, or analogy even, that gives the reader a sense for why this happens.

Response:

The lack of initial conditions (assumed usually 0) on the displacement are inconsistent with the real values that results in the phenomenon of baseline shift or drift which causes the error to grow with time during the integration [8]. The sentence along with reference [8] is added to the revised manuscript. Please refer to L 117-119, P 5.

16) Page 7, Figure 3 –There seems to be a lack of clear definition for what theta is. I assumed it was displacement of the shaft, normally taken in-line with one of the blade roots. But that definition would see theta move from 0 to 2*pi repeatedly, which is not what is happening for the 'Actual' theta in Figure 3. Now it seems that theta is instead the torsional deflection in a moving frame which rotates with the shaft. The definition needs to be made clear when the model is introduced as otherwise this sort confusion can arise. Maybe add to Figure 1 to show theta more explicitly there?

Response:

The theta refers to the torsional displacement of the shaft. It is an instantaneous change of the shaft torsional displacement about the static equilibrium position in an inertial/fixed frame of reference. The information about the frame of reference is included in the revised manuscript. Please refer to L 96-97, P 4.

17) Page 7 and 8– Mathematical presentation again. Please see previous comments. Additionally, in Equation 8 it is not clear what we are minimising over, in Equation 9 theta now denotes a vector (please use new notation when going from scalar to vector and make all definitions explicitly). These same developments include mention of 'fictitious nodes' with no description. Overall the reader is left struggling to understand what is happening and why, with little hope of being able to reproduce what is done here.

Response:

The formulation developed in Appendix A.1 of the revised manuscript (pages 7 and 8 of the submitted manuscript) is used to regularize the influence of the numerical and measurement noise on the reconstruction of the torsional displacement. This is achieved by minimizing the error between the actual and measured velocities in the least square sense as in Eq. (12). The L2-norm of the residual is being minimized in Eq. (12). In other words, Eq. (12) gives a measure of how well the actual velocity approximates the measured velocity. By following the procedure outlined in Appendix A.1, the concerned minimization problem given by Eq. (12) is reduced as a quadratic equation in Eq. (20). By using Eq. (20) with the measured velocity, one can obtain the reconstructed torsional displacement that is regularized for the numerical and measurement noise. The concerned sentences are now modified to

convey the above discussion. Please see L 124-129 in P 5 and L 285-290 in P 15 of the revised manuscript.

In the finite difference discretization of Eq. (16), some defined nodes are located outside of the domain considered (i.e., domain here is time, $t \in [t_0, t_n]$ satisfying $t_0 \le t \le t_n$). These nodes defined by time steps i=-1 and i= n+1 are fictitious. These nodes are the dummy nodes that are used in solving the differential equations by the finite difference method [9]. This definition along with reference [9] is given in the revised manuscript. Please refer to L 299-302 in Appendix A.1, P 16.

As per the reviewer's suggestion shorter descriptions of the collage method and Tikhonov regularisation along with the necessary equations and solution method is given in the problem formulation. Detailed mathematical explanations are shifted to the Appendix section. Please refer to Sections 2.1, 2.2 and Appendix of the revised manuscript. Further, all the mathematical symbols are revisited in terms of their definition and notations in the revised manuscript.

18) Page 8, line 180 – "Since ten-minute SCADA measurements with a sampling frequency of 50 Hz are considered for theta(t) estimation, n becomes 30000." Now SCADA of much higher frequency is mentioned. This is even less common that 1s data. Assumptions and requirements concerning data from the wind turbine needs much clearer description and discussion throughout the paper. Also, if 50Hz is used here, will the method suffer is that's not available? Did you test to see what you can get away with at lower frequency?

Response:

 The proposed approach requires that the sampling frequency of the SCADA measurement be significantly higher than the dominant frequencies of the drivetrain torsional oscillation. This explanation is given in the introduction (L 83-86, P 3) of the revised manuscript. (Please refer to the response of query 5).

19) Page 8, line 180 –Now thatn is becoming large it seems that a discussion of required computing power and computational times is necessary. Does this take a long time to run? How about if you were assessing a lot of data for remaining useful life analysis across a whole wind farm? I don't believe any such discussions are undertaken in the manuscript currently.

Response:

The computational time for identifying the regularisation parameter for each mean wind speed is about 40 mins in real-time and the stiffness estimation for the 10-min simulation takes 14 s in real-time. The above computations were performed on a single node of the high-performance computing cluster of

DTU. The cluster has 320 nodes in total, each node consists of two Intel Xeon E5-2680v2 processors, and each processor consists of 10 cores running at 2.8 GHz.

If the wind farm is of same type of turbine, then it is sufficient to estimate the stiffness for one of the turbines. However, the shaft torsional displacement needs to be estimated for all the turbines.

These discussions are included in the revised manuscript. Please refer to L 213-218, P 10.

> 20) Page 9, line 209 – "the inputs". Again, what frequency is this data at?

Response:

The sampling frequency of the inputs is 50 Hz. The concerned sentence is modified as follows:

The DLC 1.2 was run with eighteen 10-minute load simulations (three yaw directions: 0 deg, ± 10 deg) and six turbulent wind seeds ) for each mean wind speed and the mean wind speed is ranging from 4 m/s to 26 m/s in the interval of 2 m/s, which results in a total of 216 simulations. Each simulation is of 10-min duration sampled at 50 Hz frequency. Please refer to L 166-168, P 8.

> 21) Page 10, Figure 5 – The flow chart is blurry and appears squashed. Please remake as a clearer vector image. Also I'd include this in a 'methodology' section earlier in the paper to help make the process clearer earlier in the paper.

Response:

A new flowchart of better quality is now included in the revised manuscript and as per the reviewer's suggestion, the same is moved to the beginning of the discussion (Section 2). Please refer to Fig. 2 of the revised manuscript.

> 22) Page 10, line 216 – "At this point, it is important to realize that the static component of the displacement does not... and hence only the dynamic component of the torsional displacement" Neither of the 'static' or 'dynamic' component of torsional displacement is defined in the paper and this is the first mention that either of them get.

Response:

Since the displacement is reconstructed from the velocity- a dynamic quantity, the reconstructed displacement will have zero mean and this displacement component is referred to as a dynamic component of the displacement (theta_dyn). However, the mean torsional displacement will have a contribution from the static load and the displacement due to static load is referred to as a static

component of the displacement (i.e., the mean value of the displacement). The dynamic component of the displacement oscillates about the static component of the displacement.

These definitions are included into the revised manuscript. Please refer to L 177-181, P 8.

23) Page 11, line 219 - "By removing the static displacement from the simulated displacement, the resulting dynamic component can be compared with…". Is that useful? Is static component important for damage in the shaft? Does is relate to the mean about which fluctuations are occurring? None of this is clear and so the highlighted line of text is also unclear regarding what is being compared and how the comparison is to be interpreted.

Response:

Since the reconstructed displacement is the dynamic component of the torsional displacement, the same is compared with the dynamic component of the displacement obtained from HAWC2 simulations in Fig. 7. Upon ensuring the correctness of the reconstructed displacement, the torsional stiffness of the drivetrain is estimated by employing the collaged method. However, the mean value (i.e., the static component) of the displacement has a significant effect on the fatigue damage [10]. Hence, it is important to estimate the static component of the shaft torsional displacement. This is obtained by solving the static problem of the drivetrain. Finally, the static and dynamic components are superimposed to get the actual displacement and used to estimate the site-specific loads and fatigue damage.

The discussion is included in the revised manuscript. Please refer to L 198-209, P 9-10.

24) Page 11, line 228 – "12.06%". This sounds good but how good is it in the context of the problem at hand? How else might it be approximated and if there are other methods how does this compare?

Response:

The 12.06% deviation in estimated K from its actual value resulted in the relative error ranges from 4 % to 12 % in the computed DEL. Another way is to compare the inherent frequencies of the estimated load with the actual one as done in Fig. 7. As seen in Fig. 7, all the dominant mode frequencies are captured in the estimated load.

25) Page 11, line 228 –"At this point, it is important to remember that the estimated K value is a combination of the main shaft and blade edgewise stiffness." What does this imply for the reader? How does this change our interpretation of findings? If other components are present,

how large are they relative to shaft K. Can you perform a simple analysis be taking blade edge stiffness values from the HAWC2 model and comparing with a known value of shaft torque to demonstrate that the dominant contribution to K is from the shaft?

Response:

The drivetrain torsional frequencies of the DTU 10 MW for two different modeling assumptions (without blade edge stiffness and with it) obtained from [11] are presented here.

| | When blade flexibilities included | Rigid blade (i.e., only the main shaft is flexible) |
|---|---|---|
| Drivetrain Torsional frequency (Hz) | 1.8 | 4.003 |

As seen in the above table, the addition of blade flexibilities resulted in a 50% reduction in the drivetrain torsional frequency. This is due to the fact that the rotor inertia is significantly higher than the drivetrain components, [For DTU 10 MW turbine [11], rotor inertia, $J_r$= 8.6e07 kgm^2, generator inertia, $J_g$=3.75e06 kgm^2; all the inertia values are about the low-speed side].This is not the case for the gearbox dynamics, since the inertia of the gearbox is approximately a factor of 1 /30 of the generator inertia [4]. Hence, these components do not significantly affect the main shaft torsional frequency.

Also, as per the new modelling assumption, the concerned sentence is modified as follows:

At this point, it is important to remember that since the inputs are from the HAWC2 aeroelastic simulation which does not account for gearbox dynamics, the estimated stiffness value has the contributions from the rotor and the main shaft only. As a result, the estimated K is the resultant torsional stiffness of the main shaft including the blade edgewise stiffness contribution. Please refer to L 194-197, P 9 of the revised manuscript.

26) Page 11, line 231– Again,static and dynamic theta values need to be introduced much earlier in the paper, and described without assuming familiarity. Also the method by which static theta is estimated should form part of a methodology section, rather than being introduced in amongst results.

Response:

As per reviewer's suggestion, the definition of the static component is now be given earlier in Section 3 of the revised manuscript. Please refer to L 177-181, P 8.

*27)* Page 11, line 233 – What about gearbox efficiency? If generator efficiency is taken into account won't gearbox efficiency also have an effect? If not then please outline why, e.g. maybe the efficiency of one if much higher than the other? Otherwise both should be accounted for.

Response:

In the article, the generator efficiency is taken for the static calculations. However, as an alternative approach, one can use the overall efficiency of the drivetrain as well for the sake of completeness. This does not significantly change the results. The use of the generator efficiency resulted in the offset of 2.5 % about the mean value of the rotor torque in the estimated static component of theta. This explanation is now added to the revised manuscript. Please refer to L 203-204, P 9.

*28)* Page 12 – Damage equivalent load calculations: The Wöhler exponent takes on different values, which is used here? Does hollowness of a shaft have an effect to these calculations?

Response:

The Wöhler exponent for the cast iron of m=6 [10] is used in the study and the effect of shaft hollowness is not considered. This has been added into the revised manuscript. Please refer to L 226-227, P 11.

*29)* Page 13, line 254 – "This could be due to fact that the pitch control will be active beyond the rated wind speed and hence the instantaneous changes in the pitch angle affects the rotor speed." Isn't this a bit of a gap in the modelling? Wind turbines spend a fair amount of time operating while pitching so this is an important factor. This all ties to the approximations being made when the chosen model is applied. A better discussion of the model in relation to a real world turbine is needed when the model is presented, and this must include effects such as pitch that are accounted for, along with some indication of the possible impact that will have. E.g. "The proposed model does not account for shaft speed changes related to pitch activity, however, in above rated conditions the controller is acting to maintain a constant rotational speed, and so such changes (and associated errors) are expected to be small..." or something like that.

Response:

The fact that the peak amplitudes are not captured in the reconstructed torsional displacements is typical for Tikhonov regularisation as there is a slight mismatch in the frequency spectra between the actual and reconstructed torsional oscillations [15]. Another reason could be due to the fact that the pitch angle influences the main shaft torsional oscillation through the rotor torque and the rotor speed. However, the controller is maintaining a constant rotational speed beyond the rated wind speed, hence the instantaneous changes in the pitch angle are not propagated through the rotor speed to the main

shaft oscillations. All these effects, in addition to the uncertainty in K estimation, may result in an error of 12 % in the estimated DEL.

This explanation is now added to the revised manuscript. Please refer to L 231-237, P 11.

*30)* Page 13 (and onwards) – You start referring to "1Hz DELs" without defining what these are.

Response:

When Nref in Eq. (21) equals the component's lifetime in seconds, then the DEL has the frequency of 1 Hz. Since Nref of 600 cycles is used in the present study, it becomes 1 Hz DEL over a 10 min period. This definition is now added to the revised manuscript. Please refer to L 227-228, P 11.

*31)* Page 13, line 258 – "SCADA measurements of the Vestas V52-850 kW research turbine…" What frequency of data?

Response:

Throughout this article, the input data has a sampling frequency of 50 Hz.

The concerned sentence is modified as follows:

SCADA measurements taken during January 2019 for the Vestas V52-850 kW research turbine installed at the DTU Risø site is utilized for this purpose. The measurement data consists of 4459 ten minute recorded cases with 50 Hz sampling frequency.  Please refer to L 240-242, P 11.

*32)* Page 14, lines 278-282 – You make big claims about remaining useful life, but only actually calculate damage equivalent loads. Unless you demonstrate how this link is made (which is non-trivial) you maybe need to make the claims less strong. E.g. `the proposed methodology may assist in decision making regarding maintenance and remaining useful life' kind of thing. Or, reference to material that shows how damage equivalent load predictions are then made into remaining useful life assessments. But either way, claims need to be tentative unless backed up by references or evidence.

Response:

The estimated main shaft torsional DEL may be used to quantify the  RUL of the drivetrain [12]. However, this is beyond the scope of the current work. The concerned sentence is  modified accordingly. Please refer to L 261-262, P 13.

*33)* Page 15, line 281 – "It is also feasible to reconstruct the torsional moment time series, which may be used an input to gearbox design tools to predict the loading within the gearbox." This is another big claim, especially considering that you have assumed a perfectly stiff gearbox with 100% efficiency, if included it needs to be demonstrated more fully that this is indeed the case. It is also important to consider what you have shown, which is that damage equivalent loads are well predicted by your method, that is not necessarily the same as saying that the time-series loads are reconstructed to a similar level of accuracy. They may be, but that needs discussion and consideration and doesn't follow immediately from the damage estimate results.

Response:

As per the new modelling assumption, the estimated stiffness is the collective stiffness of the entire drivetrain since the inputs are from the measurements. Hence, the estimated torsional load can be used as an input to gearbox design tools. Please refer to L 252-254, P 12.

Regarding the reconstructed torsional load time series, all the important aspects of the time-series variation and the dominant frequency dynamics (low-frequency components - up to first three peaks ) are captured quite well in the reconstructed torsional load as compared to the simulated one as shown in Fig. 7 of the manuscript. Hence, it is established in the article that both the torsional load and the torsional DEL can be reconstructed with reasonable accuracy. However, to be consistent with the terminology, the torsional moment time series is now changed as torsional load time series.

*34)* Page 15, lines 297-298 – "As the proposed methodology does not require any design basis, it can be used for any older turbines for the estimation of the main shaft torsional RUL." This is not necessarily true. As mentioned, older turbines will at best tend to have 10min averaged SCADA data available. What does this mean for your methodology if that's the case? If high-frequency SCADA is always required then that needs to be made much clearer throughout the paper, and probably indicates that the term `high-frequency SCADA' should be used in the paper title, since most people assumed SCADA = 10min averages. The methodology is still potentially relevant to older turbines since new data can be collected to link fatigue damage to its operating conditions, but life estimates then require some sort of hindcasting or statistical analysis that goes beyond what is covered here. Again, please consider which claims are reasonable to make based on what is demonstrated in the paper, if reaching beyond what is presented here then please add more discussion and references which evidence the claims being made.

Response:

The concerned sentence has been modified based on the findings of the current work. The modified sentence reads:  As the proposed methodology does not require any design basis or an aeroelastic

design basis, it can be used for wind turbines that possess high-frequency SCADA measurements for the estimation of the main shaft torsional load and DEL. Please refer to L 278-280, P 14.

Please refer to the previous discussions (i.e., response to query 5) about the sampling frequency of the SCADA measurement. In addition, the phrase 'high-frequency SCADA' is now included in the title as per the reviewer's suggestion.

**References:**

[1] Boukhezzar, B., Lupu, L., Siguerdidjane, H., & Hand, M. (2007). Multivariable control strategy for variable speed, variable pitch wind turbines. Renewable energy, 32(8), 1273-1287.

[2] Shin, Y. H., Moon, S. J., Kwon, J. I., & Chung, T. Y. (2013). Derivation of 4 degrees of freedom nonlinear wind turbine model using effective mass and stiffness for simulation of control algorithm. Journal of Renewable and Sustainable Energy, 5(5), 052012.

[3] Berglind, J. B., Wisniewski, R., & Soltani, M. (2015, July). Fatigue load modeling and control for wind turbines based on hysteresis operators. In 2015 American Control Conference (ACC) (pp. 3721-3727). IEEE.

[4] Novak, P., Ekelund, T., Jovik, I., & Schmidtbauer, B. (1995). Modeling and control of variable-speed wind-turbine drive-system dynamics. IEEE Control Systems Magazine, 15(4), 28-38.

[5] Singh, M., & Santoso, S. (2011). Dynamic models for wind turbines and wind power plants (No. NREL/SR-5500-52780). National Renewable Energy Lab. (NREL), Golden, CO (United States).

[6] Girsang, I. P., Dhupia, J. S., Muljadi, E., Singh, M., & Pao, L. Y. (2014). Gearbox and drivetrain models to study dynamic effects of modern wind turbines. IEEE Transactions on Industry Applications, 50(6), 3777-3786.

[7] Pahn, T. (2013). Inverse load calculation for offshore wind turbines (Doctoral dissertation, Hannover: Gottfried Wilhelm Leibniz Universität Hannover).

[8] Pahn, T., Rolfes, R., & Jonkman, J. (2017). Inverse load calculation procedure for offshore wind turbines and application to a 5-MW wind turbine support structure. Wind Energy, 20(7), 1171-1186.

[9] Lapidus, L. and Pinder, G. F., Numerical Solution of Partial Differential Equations in Science and Engineering], John Wiley & Sons, New York (1982).

[10] Veldkamp, H. F. (2006).  Chances in wind energy: A probalistic approach to wind turbine fatigue design (Doctoral dissertation, Faculty of Mechanical Maritime and Materials Engineering, TU Delft).

[11] Bak, C., Zahle, F., Bitsche, R., Kim, T., Yde, A., Henriksen, L.C., Hansen, M.H., Blasques, J.P.A.A., Gaunaa, M. & Natarajan, A., 2013. The DTU 10-MW reference wind turbine. In Danish Wind Power Research 2013.

[12] Pagitsch, M., Jacobs, G., & Bosse, D. (2020). Remaining Useful Life Determination for Wind Turbines. In Journal of Physics: Conference Series (Vol. 1452, No. 1, p. 012052). IOP Publishing.

[13] A Brandta and R. Brincker, Integrating time signals in frequency domain – Comparison with time domain integration, Measurement 58 (2014) 511-519.

[14] Qihe, L. (2019, November). Integration of vibration acceleration signal based on labview. In Journal of physics: conference series (Vol. 1345, No. 4, p. 042067). IOP Publishing.

[15] Lee, H. S., Hong, Y. H., & Park, H. W. (2010). Design of an FIR filter for the displacement reconstruction using measured acceleration in low-frequency dominant structures. International Journal for Numerical Methods in Engineering, 82(4), 403-434.

**Reviewer 2**

**General comments**

1) The first question concerns the simplified mathematical model of the drivetrain (Eqs 1-3). This model is a compact two-disk representation, in which the inertia of the main shaft and that of the gear box and high-speed shaft are collected into the equivalent generator inertia. Therefore, for the sake of consistency of the model, the stiffness term (K) seems to stand for the collective stiffness of the whole drivetrain rather than solely that of the main shaft [1]. However, the manuscript is taking whole drivetrain apart from the main shaft perfectly rigid, so above stiffness is assigned to the main shaft and will then be identified and utilised it to infer the main shaft loads, which may not be right for drivetrains with multi-stage gearbox and shafts. There are several references available (e.g. [2]) that explain the equivalent drivetrain models as the combination of springs in series. The angular displacement of such models are basically the summation of the individual displacements of the drivetrain components. A comparison with simulation results cannot simply validate the above assumption, as the turbine simulation tools are also usually including a simplified drivetrain model, as they aim to capture the global motion and loads of system. In case that the contribution of the other components are known/found to be negligible, this has to be quantitatively demonstrated or at least be discussed adequately. It is not well described whether this work is identifying the main shaft load or the total equivalent torsional load of the drivetrain. Nevertheless, the manuscript's methodology and findings are still interesting. As such, it is recommended to either clarify this point or discuss the probable limitations.

Response:

The two-mass model is widely used to study the torsional oscillations of the mechanical drivetrain of a wind turbine [1-5] because it captures the underlying free-free torsional frequency. This is the main driving frequency for the main shaft torsion. Though the functional form of the governing equations (i.e., Eqs. (1-3) of the manuscript) remain the same for all the considered work, the underlying assumption regarding the stiffness of the various components in the drivetrain is different for each model and the definition for the 'K' in Eqs. (1-3) in each of these works is given in the following table:

| Model | Considered stiffnesses for K |
|---|---|
| Boukhezzar et al. [1] | Rotor, generator and the main shaft |
| Shin et al. [2] | Main shaft |
| Berglind et al. [3] | Collective stiffness with rigid gearbox |
| Novak et al., [4] | Main shaft |

| Singh & Santoso, [5] | Main shaft, high-speed shaft and gearbox dynamics neglected |
|---|---|
| Girsang et at., [6] | Stiffness of all components were considered. |

As seen in the table, the modelling assumptions regarding the gearbox and main shaft are inspired from [1-3]. However, for the inverse-problem based approach, the inputs to the drivetrain model (i.e., the rotor and generator speed) determine the contribution of each component to the 'K' value. For example, if the inputs are from the HAWC2 aeroelastic simulation then the estimated stiffness value has the contributions from the rotor and the main shaft only since the simulation does not account for the gearbox dynamics. If the inputs from the measurements are used, then the estimated stiffness is a collective stiffness that has a contribution from all the components.

To be consistent with all the inputs, the modelling assumptions adopted by Girsang et al.[6] has been referred in the revised manuscript (i.e., by considering the contributions from all the components in the drivetrain). Nevertheless, these modelling assumptions will not affect the methodology proposed in the manuscript.

The concerned sentences (i.e., page 3, line 86 and line 88 and Page 4, line 98 and 99 of the reviewed manuscript) is removed in the revised manuscript. Accordingly, the discussion about the inputs to the model is included along with the reference (Girsang et al., [6]) in the revised manuscript. Please refer to L 96-99 and L 103-105, P 4 of the revised manuscript.

2)  The other point is regarding the manuscript's argument that it just needs SCADA data, which sounds compelling due to this data being readily available. The suitable sampling rate of the SCADA data that is required for this methodology has not been discussed within the paper and needs to be closely clarified. This point becomes more important when the manuscript argues that their findings are beneficial to the calculations for life extension of wind turbine system. However, many of the existing turbines that would require life extension are equipped with SCADA systems with very low frequency output data, normally averaged values of order of minutes. It seems that this drawback is going to be removed by artificially increasing the SCADA signal sampling rate without providing any information on the original and the resulted sampling rate. Therefore, it is a far question whether this methodology is really applicable to SCADA data or condition monitoring system's data is still required. As such, it is again recommended to either clarify this point or discuss the probable limitations.

Response:

We thank the reviewer for pointing out this fact. The proposed approach requires that the sampling frequency of the SCADA measurement be significantly higher than the dominant frequencies of the drivetrain torsional oscillations (i.e., 1p and 3p rotor excitation frequencies and torsional natural frequencies). As a result, the proposed method cannot be used for the turbines that have measurements in terms of 10-min SCADA statistics. In addition, we have used the phrase high-frequency SCADA in the article title as well. Also, the authors are neither proposing any method for the new data collection nor resampling the measurement data. It is demonstrated that with the existing high-frequency SCADA measurements, the site-specific torsional load can be estimated. This can be used to estimate the yearly damage without historical weather records or condition monitoring data as the wind speed and wind direction measurements are available in the SCADA measurements.

Also, the arguments regarding the estimation of the RUL have also changed to emphasize that those are not the scope of the current work.

We have rewritten the concerned paragraph of the revised manuscript to convey the about discussion. Please refer to L 81-86, P 3 of the revised manuscript.

3) Moreover, there are well-established integration techniques that are being used, even for online conversion of acceleration to velocity and displacement signals [3, 4]. In the absence of artificial noise in the simulated speed data, the type of drift shown in Figures 3 and 4 are apparently showing a linear trend due to the accumulation of the error from the initial value. This type of trend can be usually avoidable by the common digital filtering with restriction of frequency range within the pre-processing of the signal, particularly when the signal's mean value (static term) is going to be added later on separately, in which case the initial value of integration doesn't really matter. As such, the authors need to mention that the application of regularisation within a trapezoidal scheme is not the only available method to stably convert velocity into displacement.

Response:

Digital filters and frequency domain integration approach (FDIA) are the widely used techniques in literature to reconstruct displacements from the measured accelerations [7-9]. However, digital filters such as impulse response filters (IIR) and finite response filters (FIR) have several drawbacks when reconstructing the low-frequency displacements, as is the case here [9]. On the other hand, the FIDA methods are sensitive to the time intervals of the measurements [9]. Hence, a least square minimization based regularization technique called Tikhonov regularisation is used in the present study as it is better suited for low-frequency dominant structures as shown by [9].

The above discussion is added into the introduction of the revised manuscript. Please refer to L 58-62.

**Minor comments:**

1) The derivatives for the inverse integration scheme is similarly given in Hong et al 2008, so please refer to this paper in the beginning of the corresponding section.

Response:

The concerned reference has now been added into the revised manuscript. Please refer to L 47-48 and L 56-58.

2) Paragraph 15: "and" missing after strain gauges.

Response:

The reviewer's suggestion will be incorporated into the revised manuscript. Please refer to L 28, P 2.

3) Paragraph 75: This is questionable if older turbines really possess SCADA with a desirable sampling rate that suits the manuscript's methodology.

Response:

Please refer to the response of the general comments 2.

4) Paragraph 85: As discussed previously, the assumption of the fully rigid gearbox and higher stage shafts, particularly for a multi-stage systems, needs further clarifications.

Response:

Please refer to the response of the general comments 1.

5) Paragraph 190: computationally "more" expensive.

Response:

The reviewer's suggestion is incorporated into the revised manuscript. Please refer to L 319, P 16.

6) Paragraph 205: Please clarify those three yaw directions.

Response:

The reviewer's suggestion is incorporated into the revised manuscript. Please refer to L 166, P 8.

7) Paragraph 210: where is the noise coming from that needs to be damped. Please clarify.

Response:

The concerned place talks about the numerical noise that arise due to the lack of initial conditions. Please refer to L 172-173, P 8 of the revised manuscript.

8) Paragraph 220: "the" modes.

Response:

The reviewer's suggestion is incorporated into the revised manuscript. Please refer to L 192, P 9.

9) Paragraph 225: pds should be expanded.

Response:

The explanation for pdf (i.e, probability density functions)  is now given in the revised manuscript. Please refer to L 193, P 9.

10) Mixed use of symbols has to be avoided throughout the manuscript.

Response:

The reviewer's suggestion is incorporated into the revised manuscript and mixed symbols have been avoided throughout the revised manuscript.

11) Eq 18, it seems that gearbox ratio is missing somewhere.

Response:

In the revised manuscript, all quantities from the HSS side have been transferred to the low-speed side of the drivetrain by using the gear ratio. This discussion has now been added in the revised manuscript. Please refer to L 96-97, P 4.  As per this, all the quantities presented in the revised manuscript are on the low-speed side.

12) Fig 8, it seems that some constant or near constant offset exist (observable in both time and frequency plots), apparently theta_static is not appropriately identified. The plotted FFT is too noisy, one can use a better illustration, perhaps, a power spectrum with sufficient averaging to

get more clear peaks, as at the moment it is barely possible to get a good picture of the main peaks.

Response:

The constant offset is due to the use of the generator efficiency for all the static displacement calculations. As a result, there is an offset of 2.5 % about the mean value of the rotor torque. However, as an alternative approach, one can use the overall efficiency of the drivetrain as well which accounts for the efficiencies of all components for completeness.

 Power spectral density (PSD) plots are given in the revised manuscript instead of the FFT plots.

13) Paragraph 20: what are the sampling rate details and how the SCADA data's frequency resolution was increased, is it practically legitimate? Please clarify.

Response:

The sampling frequency of the SCADA measurement is 50 Hz. In the considered work, the authors have not increased the sampling rate of the measurement.

14) The results using the actual data does not look to add substantial value to the manuscript, as it is not really supporting the validation of the methodology.

Response:

The authors feel that including the results from the V52 measurements shows that the resulting trend is very similar to the trend observed from the HAWC2 results. Since RUL prediction on wind farms is based on the margins available on the actual structure as compared to prior aeroelastic computations, this result demonstrates the confidence in the proposed method to be utilized in RUL computations of wind farms.

**References:**

[1] Boukhezzar, B., Lupu, L., Siguerdidjane, H., & Hand, M. (2007). Multivariable control strategy for variable speed, variable pitch wind turbines. Renewable energy, 32(8), 1273-1287.

[2] Shin, Y. H., Moon, S. J., Kwon, J. I., & Chung, T. Y. (2013). Derivation of 4 degrees of freedom nonlinear wind turbine model using effective mass and stiffness for simulation of control algorithm. Journal of Renewable and Sustainable Energy, 5(5), 052012.

[3] Berglind, J. B., Wisniewski, R., & Soltani, M. (2015, July). Fatigue load modeling and control for wind turbines based on hysteresis operators. In 2015 American Control Conference (ACC) (pp. 3721-3727). IEEE.

[4] Novak, P., Ekelund, T., Jovik, I., & Schmidtbauer, B. (1995). Modeling and control of variable-speed wind-turbine drive-system dynamics. IEEE Control Systems Magazine, 15(4), 28-38.

[5] Singh, M., & Santoso, S. (2011). Dynamic models for wind turbines and wind power plants (No. NREL/SR-5500-52780). National Renewable Energy Lab. (NREL), Golden, CO (United States).

[6] Girsang, I. P., Dhupia, J. S., Muljadi, E., Singh, M., & Pao, L. Y. (2014). Gearbox and drivetrain models to study dynamic effects of modern wind turbines. IEEE Transactions on Industry Applications, 50(6), 3777-3786.

[7] A Brandta and R. Brincker, Integrating time signals in frequency domain – Comparison with time domain integration, Measurement 58 (2014) 511-519.

[8] Qihe, L. (2019, November). Integration of vibration acceleration signal based on labview. In Journal of physics: conference series (Vol. 1345, No. 4, p. 042067). IOP Publishing.

[9] Lee, H. S., Hong, Y. H., & Park, H. W. (2010). Design of an FIR filter for the displacement reconstruction using measured acceleration in low-frequency dominant structures. International Journal for Numerical Methods in Engineering, 82(4), 403-434.